# A dynamic attractor network model of memory formation, reinforcement and forgetting

**Marta Boscaglia**[1,2]*, **Chiara Gastaldi**[3], **Wulfram Gerstner**[3], **Rodrigo Quian Quiroga**[1,4,5,6]*

**1** Centre for Systems Neuroscience, University of Leicester, United Kingdom, **2** School of Psychology and Vision Sciences, University of Leicester, United Kingdom, **3** School of Computer and Communication Sciences and School of Life Sciences, École Polytechnique Fédérale de Lausanne (EPFL), Switzerland, **4** Hospital del Mar Medical Research Institute (IMIM), Barcelona, Spain, **5** Institució Catalana de Recerca i Estudis Avançats (ICREA), Barcelona, Spain, **6** Ruijin hospital, Shanghai Jiao Tong University School of Medicine, Shanghai, People's Republic of China

* mb899@le.ac.uk (MB); rqqg1@le.ac.uk (RQQ)

## Abstract

Empirical evidence shows that memories that are frequently revisited are easy to recall, and that familiar items involve larger hippocampal representations than less familiar ones. In line with these observations, here we develop a modelling approach to provide a mechanistic understanding of how hippocampal neural assemblies evolve differently, depending on the frequency of presentation of the stimuli. For this, we added an online Hebbian learning rule, background firing activity, neural adaptation and heterosynaptic plasticity to a rate attractor network model, thus creating dynamic memory representations that can persist, increase or fade according to the frequency of presentation of the corresponding memory patterns. Specifically, we show that a dynamic interplay between Hebbian learning and background firing activity can explain the relationship between the memory assembly sizes and their frequency of stimulation. Frequently stimulated assemblies increase their size independently from each other (i.e. creating orthogonal representations that do not share neurons, thus avoiding interference). Importantly, connections between neurons of assemblies that are not further stimulated become labile so that these neurons can be recruited by other assemblies, providing a neuronal mechanism of forgetting.

## Author summary

Experimental evidence suggests that familiar items are represented by larger hippocampal neuronal assemblies than less familiar ones. In line with this finding, our computational model shows that the size of memory assemblies depends on the frequency of their recall (i.e. the higher the frequency, the larger the assembly), which can be explained by the interplay of online learning and background firing activity. Furthermore, we find that assemblies representing uncorrelated memories increase their sizes while remaining orthogonal, in line with findings with single-cell recordings. To model these empirical

**Data Availability Statement:** All relevant data are available on figshare at https://doi.org/10.25392/leicester.data.24786912.v1 and the code used to produce the results described in the manuscript is

available on GitHub at https://github.com/
MartaBoscaglia/DynamicAttractorNetworkModel_
2023.

**Funding:** RQQ and MB were supported by the
Biotechnology and Biological Sciences Research
Council (https://www.ukri.org/councils/bbsrc/),
grant reference number BB/T001291/1. WG and
CG were supported by the Swiss National Science
Foundation (https://www.snf.ch/en), grant
agreement 200020_184615 and by the European
Union Horizon 2020 Framework Program (https://
ec.europa.eu/programmes/horizon2020/) under
agreement no. 785907 (HumanBrain Project,
SGA2). The funders had no role in study design,
data collection and analysis, decision to publish, or
preparation of the manuscript.

**Competing interests:** The authors have declared
that no competing interests exist.

findings, we propose to go beyond the standard attractor network memory models and
use instead a dynamic model to study memory coding.

## Introduction

Memories are continuously shaped by our experiences [1]. When meeting a new person, we
build a new memory that can be reinforced if meeting the same person again several times.
Alternatively, if we do not meet this person again, or if too much time passes until the next
encounter, we may forget them. This simple example illustrates that memories are dynamic
and dependent on how often they are revisited. While it has been argued that the efficiency of
a memory system is measured through the stability of its representations through time, it is in
fact the dynamical changes of memory representations that support flexible behaviour and
generalization [2]. In other words, our memories are far from static engravings in a wax tablet,
as famously argued by Plato, and we should consider how they can be modulated depending
on further experiences. In fact, it seems natural to reinforce the representations of people that
become familiar after several encounters and not of others that we do not meet again.

At the neuronal level, memories are represented by specific patterns of neural activity [3,4]
and each time a memory is recalled, the strength of the synaptic connections among the neu-
rons forming the corresponding memory trace tends to be enhanced [5–7]. Conversely, the
strength of the connections within ensembles of neurons encoding memories which are not
frequently revisited tends to fade [8–10].

Following the seminal study of patient H.M. [11], it has long been recognized that the hip-
pocampus and surrounding cortex, known as the medial temporal lobe (MTL), is critical for
memory functions [12,13]. Neurons in the MTL show highly selective, invariant and explicit
responses to specific people or objects [14,15], and, when measuring responses to individual
objects or people, there is a larger tendency to find responses to familiar things [16]. Conse-
quently, there is a relationship between the familiarity of a stimulus and the size of the assem-
bly encoding it–i.e. the more familiar the stimulus, the larger the assembly representing it and,
therefore, the higher the chance to find neurons responding to familiar stimuli compared to
those coding for relatively unknown items. This is in line with fMRI results showing a larger
hippocampal activation for stimuli that are known to the subjects in a visual paradigm [17].
The larger hippocampal activation for familiar memories has been proposed to act as a booster
for their recollection [17], contributing to the fact that people tend to better recall things that
are personally relevant to them [18–20].

Memory storage and recall in the hippocampus have been modelled using attractor neural
networks [4,21–25]. In attractor neural networks each memory (or pattern) is stored in a set of
neurons that gets activated via pattern completion, whenever a subset of them receives a large
enough input. Memory patterns are encoded by assemblies of neurons strongly connected to
each other which are formed via Hebbian learning—i.e. neurons that tend to be coactivated by
an input increase their synaptic connections [7]. Nevertheless, the standard design of attractor
models [4] only reflects the final outcome of this Hebbian learning process and, after an initial
learning phase, the synaptic connectivity is fixed and memories become static attractors repre-
sented by assemblies that do not change anymore. This static configuration after learning does
not capture the dynamic nature of real-life memory processes, involving memory formation,
reinforcement and forgetting. Attractor models using dynamic learning rules have been pro-
posed [26–37]. However, only few of these studies have implemented Hebbian learning in a
way that memory assemblies can be continuously formed and dynamically updated [33–37].

These studies particularly focused on the analysis of the compensatory mechanisms needed to implement Hebbian learning in a stable manner but none of them has investigated how memory representations could change with the familiarity of the stimulus–i.e. updating the assemblies storing the memories depending on how often the memories are revisited.

In this work, we designed a rate attractor neural network with dynamic synapses to model the processes of memory formation, reinforcement and forgetting and used this model to replicate the experimental finding that the size of the assembly should increase with the frequency of presentation of the stimulus, as found with human MTL neurons [16]. Fig 1A illustrates our main hypothesis. At $t_0$, the network is at rest and there are only weak connections between the neurons. At $t_1$, a group of neurons is stimulated, which leads to the increase of their connection strength, eventually forming a neural assembly. At $t_N$, further stimulation brings the connections within the assembly to their maximal values. Furthermore, due to spontaneous activity, weak connections form between the assembly neurons and other neurons that were not stimulated (in light grey) but happened to fire by chance during the presentation of the stimulus. These weak connections increase the probability of these neurons to fire when the stimulus is presented again. When they do so, they reinforce their connections with the assembly neurons and may become part of the assembly, thus firing at a later time $t_{N+M}$ consistently with each stimulus presentation, even if not directly stimulated, by pattern completion. Conversely, those neurons that initially formed weak connections with the assembly and do not fire again at the time of the new stimulation go back to baseline connections' levels and will not join the assembly.

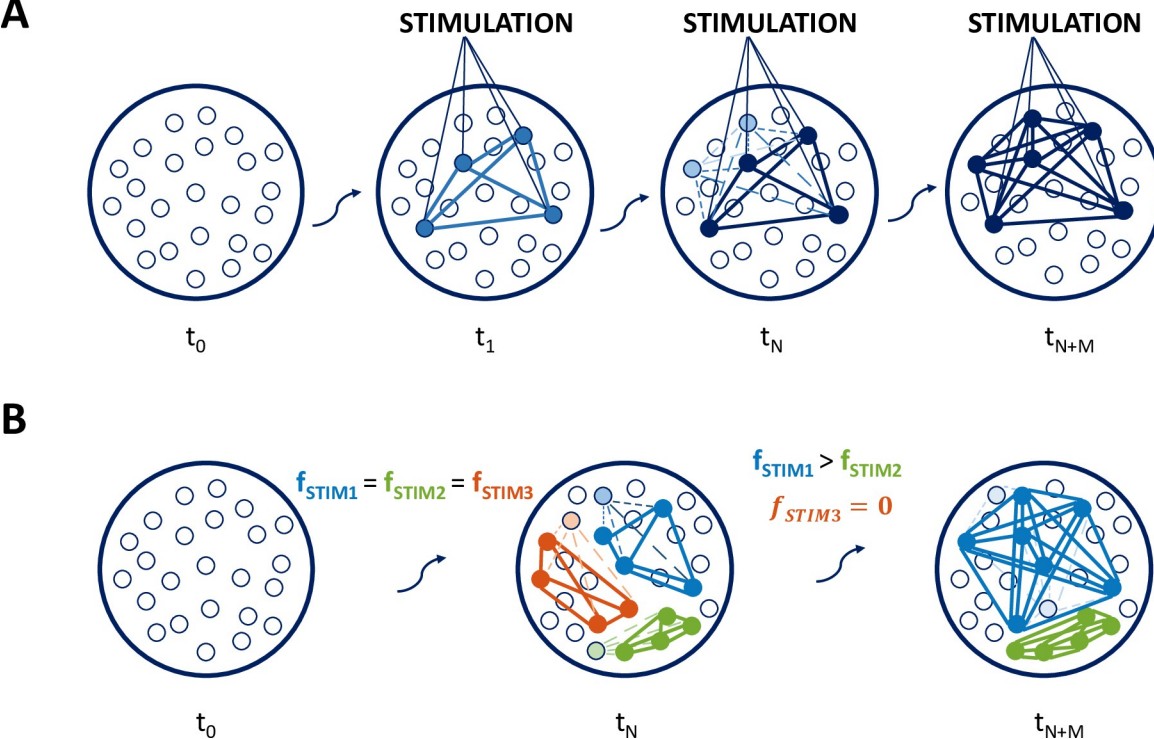

**Fig 1. Hypothesis of interplay between Hebbian learning and background noise in assembly formation and evolution.** A) Schematic of assembly formation and evolution for a network storing and retrieving one assembly. At time $t_{N+M}$ two further neurons have joined the assembly. B) Formation and evolution of multiple non-overlapping assemblies. The 3 assemblies undergo the same assembly formation phase ($t_0 \ldots t_N$) but, during the following stimulation phase ($t_N \ldots t_{N+M}$), the assembly evolution is different for the 3 assemblies. $f_{stim1} \ldots f_{stimN}$ denote the repetition frequency of stimulation.

An obvious consequence of the mechanism represented in Fig 1A is that the size of an assembly depends on how often it is stimulated, as illustrated in Fig 1B. We therefore hypothesize that assemblies that are not stimulated will tend to disappear, while those that are repeatedly stimulated will tend to grow proportionally to the frequency of stimulation (Fig 1B). In order to model this mechanism, we designed a network starting from the rate attractor model presented in Gastaldi et al. [38], introducing an online Hebbian learning rule and noisy firing rate activity, to study how the size of assemblies changes depending upon the presentation of new and familiar stimuli.

## Results

We modelled the joint action of Hebbian learning [7], with ongoing neural activity [36,39,40] and repeated stimulus presentation to study whether the size of the assemblies representing memories in the MTL depends on the familiarity of the stimulus [16]. We built our model starting from a classic recurrent neural network of N rate neurons with fixed firing threshold, fixed synapses' configuration and no background firing activity [4], which we adapted to have dynamic memory representations, as described in Methods and below. First, we introduced a neural adaptation mechanism through the use of a moving firing threshold $\theta_i(t)$ [41–44]. Next, we introduced ongoing background activity (Eq 6) and Hebbian learning (Eq 7), such that, after repeated stimulation of a pattern, the connections among neurons with correlated activity (those encoding the pattern) were selectively strengthened.

We tested how the combined action of Hebbian learning and background activity influences the network connectivity, using a network of N = 100 neurons, where 10 of them were stimulated with a current of duration T = 5 a.u. and intensity $I_0 = 1$ for 15 times (Fig 2A, left). Before the beginning of the simulation, connections were initialized at zero. When the top 10 neurons were stimulated, a weak assembly was created (at $t_1$), which got reinforced after 14 repetitions of the stimulation ($t_{15}$). We found that at $t_{15}$ new connections were formed between stimulated neurons and non-stimulated neurons. This was due to some of the non-assembly neurons firing by chance during the initial presentations of the stimulus, therefore establishing an initially weak connection with the assembly that got reinforced if they happened to fire again during the following presentations. Thus, we observe at $t_{15}$ that two of the neurons (number 19 and 98), which were initially not part of the assembly, have established a strong connection with the stimulated neurons, thus joining the assembly. In contrast, when the same stimulation paradigm was applied for the model with online learning but without noise (Fig 2A, right), no connections were formed between the stimulated and non-stimulated neurons at $t_{15}$, which means that, as expected, both the noise and the learning rule are necessary to have dynamically changing assemblies.

We also introduced an ongoing decay of synaptic strengths. Without synaptic decay, the standard deviation of the connection matrix increases drastically, since individual weights can grow to their maximal values (see Fig 2B). The decay of synaptic connections is consistent with the idea that memories tend to be forgotten if not frequently recalled [2,8–10,45,46].

Finally, we introduced two factors, $S_{R,i}(t)$ and $S_{W,i}(t)$, in order to mimic the effect of additional plasticity mechanisms. The introduction of those factors was necessary in order to have a functional behaviour of the network, as shown in Fig 2C with a network of N = 100 units subject to a rectangular pulse train stimulation. Without $S_{R,i}(t)$ and $S_{W,i}(t)$ (upper panel of Fig 2C), when new neurons were added to the stimulated assembly ($t_{55}$), the assembly activity did not go down to baseline firing after the stimulation offset (t = 51625 a.u.). This is because, following the recruitment of new neurons, the recurrent input into the assembly was so strong that it always stayed above the moving threshold (i.e. neural adaptation stopped working

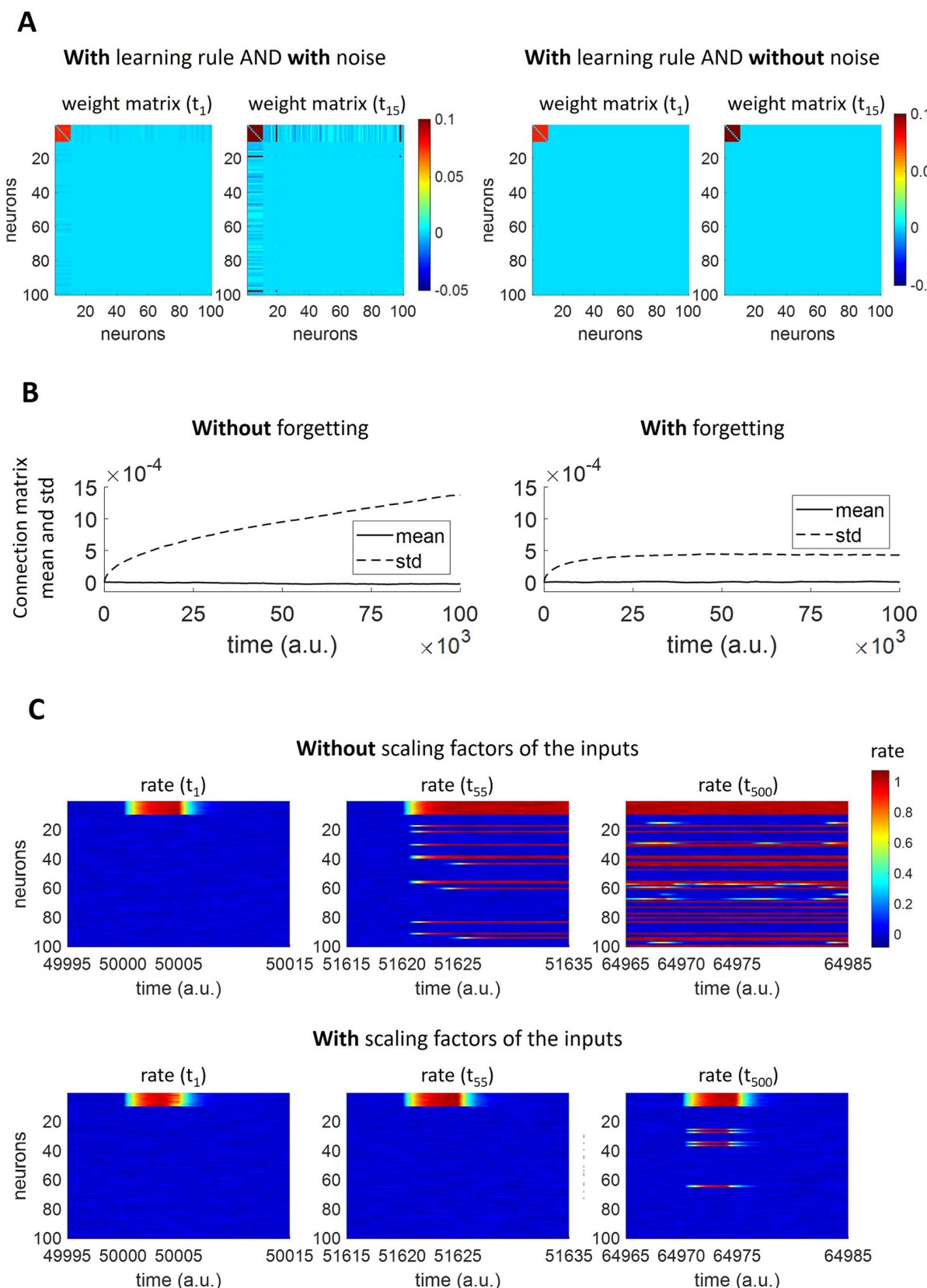

**Fig 2. Model construction.** A) Role of spontaneous background activity and learning rule. Model with learning rule and with noise (left). Model with learning rule and without noise (right). (Parameters: $\alpha_w = 0$; $\alpha_r = 0$; $w_{max} = 0.1$; $w_{min} = -0.05$; $D_\theta = 1$; $\tau_\theta = 7$ $a.u.$; $\eta = 1$; $\tau_w = 50$ $a.u.$; $\beta = 0$; $C = 0$ and $C = 0.006$ for the cases without and with noise, respectively B) Mean and standard deviation of the connection matrix, without and with forgetting. The mean and standard deviation of all network connections in absence of any external stimulation are displayed both in the case of model without (i.e. $\beta = 0$; left) and with (right) forgetting. (Parameters: $\eta = 1$; $\tau_w = 50$ $a.u.$; $\alpha_w = 0$; $\alpha_r = 0$; $w_{max} = 0.1$; $w_{min} = -0.05$; $D_\theta = 1$; $\tau_\theta = 7$ $a.u.$; $C = 0.006$; $\beta = 0$ and $\beta = 0.0025$ for the cases without and with forgetting, respectively.) C) Firing rates for all network neurons at different times, without and with scaling factors of the inputs. (Parameters: $\alpha_w = 0$; $\alpha_r = 0$; $w_{max} = 0.1$; $w_{min} = -0.05$; $D_\theta = 1$; $\tau_\theta = 7$ $a.u.$; $C = 0.006$; $\eta = 1$; $\tau_w = 50$ $a.u.$; $\beta = 0.0025$; $f = 1/(30 a.u.)$. In case of scaling factors of the inputs: $\alpha_w = 1$; $\alpha_r = 2$; $w_{max} = 0.3$; $w_{thr} = 0.05$.).

effectively). Conversely, when $S_{R,i}(t)$ and $S_{W,i}(t)$ were appropriately implemented (lower panel of Fig 2C), those factors counterbalanced the increase in recurrent input. Indeed, in that case, when new neurons were recruited ($t_{500}$), the assembly activity went down to baseline levels after the offset of the external stimulation (t = 64975 a.u).

The network had different mechanisms acting on different timescales. This is highlighted in Fig 3, where 10 neurons in a network of N = 100 were stimulated 10 times, beginning at t = 50000 a.u. The forgetting acts on a slower timescale (represented by the decay of the mean weight within the assembly, shown in the lower panel) than the memory formation and reinforcement (represented by the increase of the mean weight, upper-left panel), which in turn act on slower timescales than the neural activation and adaptation (represented by the increase of rate and firing threshold, respectively, upper-right panel). In the upper-right panel, it can be observed that the dynamics of the firing threshold (regulated by the neural adaptation mechanism) follows the dynamics of the neural activation, eventually leading to the decay of neural activation after the offset of the external stimulation (t = 50275 a.u.).

## Assembly formation and reinforcement

First, we studied the dynamics of memory formation and reinforcement using a single pattern. For this, we repeatedly stimulated 10 neurons, within a network of N = 100, with rectangular pulses. As observed in Fig 4A, the mean weight within the stimulated neurons initially increased during each stimulation and reached the maximum value after 5 stimulations. The response of one of the stimulated neurons to test stimulations (see Experimental Paradigms in

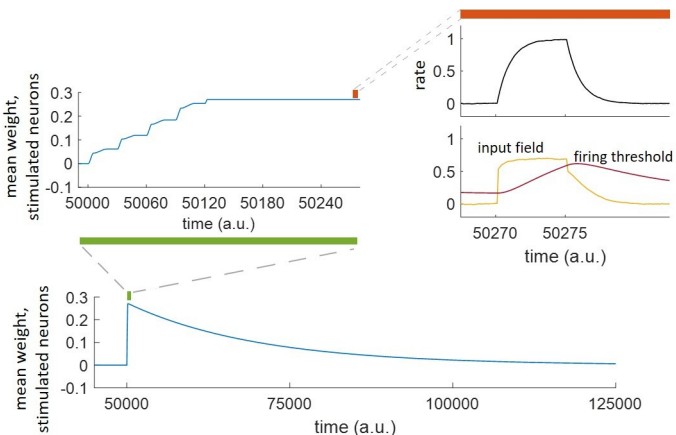

**Fig 3. Timescales of forgetting, learning, neural adaptation and activation.** Starting at time 50000 $a.u.$, a specific population was stimulated repeatedly 10 times with a repetition frequency $f = \frac{1}{30\ a.u.}$. The weight increased during the first 130 time steps after the beginning of the stimulation and decayed at the end of the stimulation ($t = 50275$ $a.u.$) over about 50000 time steps. The mean weight, rate, input field and firing threshold of the stimulated neurons are shown, zooming at timescales of interest.

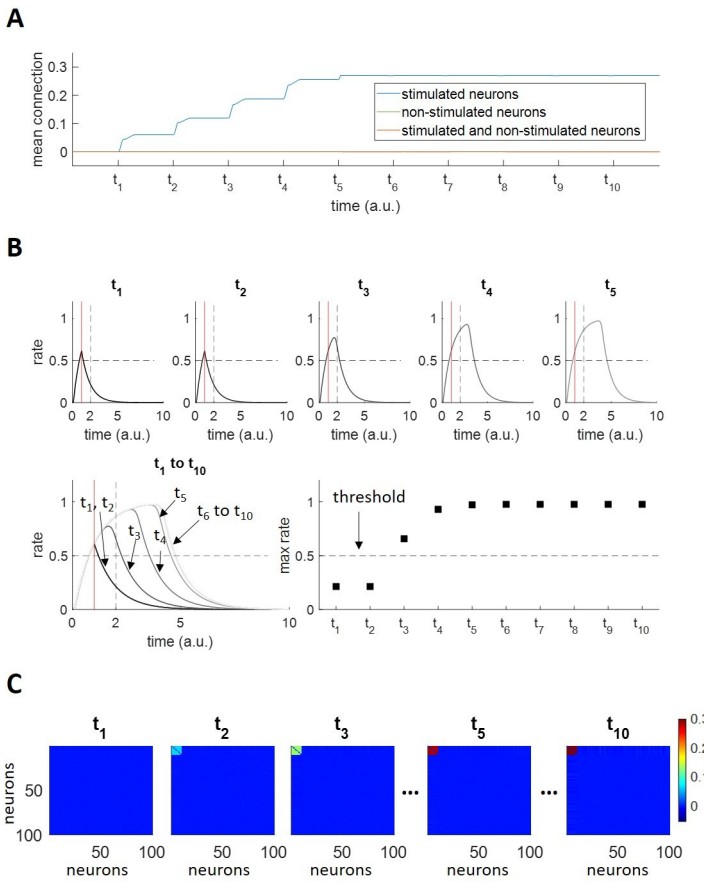

**Fig 4. Assembly formation and reinforcement.** A) Mean synaptic connection, within stimulated neurons (blue), within non-stimulated neurons (green) and among stimulated and non-stimulated neurons (orange). $t_1$ to $t_{10}$ indicate the stimulations' onset times. The mean connection within stimulated neurons increases upon stimulation, since their firing rates are strongly correlated, until the hard upper bound of the weights is reached. The curves depicting the mean connection within non-stimulated neurons and the mean connection among stimulated and non-stimulated neurons are superimposed since their values remain close to zero. B) Responsiveness criterion for an exemplar stimulated neuron. The activity traces of one of the stimulated neurons when tested at $t_1$ to $t_{10}$ are displayed. Within each plot: the vertical red line indicates the end of the test stimulation ($T = 1$ *a.u.*); the horizontal dashed black line refers to the threshold used to define a significant reverberating activation. The first reverberating activity (i.e. the attractor is about to be formed) is observed at $t_3$; from $t_4$ to $t_{10}$ the maximum value of the firing activity increases (i.e. the attractor has been reinforced) until it stabilizes. C) Weight matrices at times $t_1$, $t_2$, $t_3$, $t_5$, $t_{10}$.

Methods for further details on test stimulations) at each stimulation onset ($t_1, t_2 \ldots t_{10}$) is shown in Fig 4B. After the first two stimulations, we observe a transient response that decayed to zero immediately after the stimulation offset (marked with the red vertical line). Then, for the third stimulation, we observe that the transient response kept increasing further after the stimulation offset, due to the recurrent inputs coming from the other stimulated neurons, which could temporarily sustain the activation because of the increased weights shown in Fig 4A. This transient response increased further with further stimulations, saturating after 5 stimulations, because the connections between the stimulated neurons had reached the maximum weight. In order to establish whether a neuron was part of an assembly, we considered the firing rate value at twice the stimulation time (t = 2 a.u.) and compared it with a threshold, set at $r_{thr} = 0.5$ (see Methods). Consistent with the previous arguments, we observe that at $t_3$ the neuron had joined the (newly formed) assembly (Fig 4B, bottom right panel).

The process of the assembly formation due to repeated stimulation of a subset of neurons can also be seen in the connectivity matrix shown in Fig 4C, where we observe an increase in the connection strengths for the first 10 neurons after each stimulation, reaching a maximum connectivity after 5 stimulations. Although it is very difficult to objectively define biological or behavioural time scales, as these depend on many factors (e.g. emotional saliency, context, association with other memories), it can be noted that the formation and reinforcement of the assembly within 5 repetitions of the stimulation seems reasonable—i.e. not saturating at the first presentation of the stimulus, thus showing the effect of reinforcing the assembly connections, but also not taking way too many trials.

## Assembly evolution

Next, we characterized the evolution of an assembly that gets further stimulated following its formation and reinforcement. As before, we repeatedly stimulated 10 out of N = 100 neurons, but in this case 7000 times.

The firing of the neurons and their connections changed during the simulation (Fig 5A). In the first stage, at 1020 a.u. (after 17 stimulations), we observe that the stimulation activates the stimulated assembly, as expected. In the connectivity matrix, there are strong connections between the stimulated neurons, which had already created a strong assembly, following the process shown in the previous section. Importantly, no connections had formed between the assembly neurons and the other neurons. In the second stage, at 150,000 a.u. (after 2500 stimulations), both the firing rate plot and the connectivity matrix show that the stimulated assembly had recruited 15 other neurons, in line with the hypothesis that repeated stimulation leads to an increase of the assembly representing the stimulus (see Fig 1). At 300,000 a.u. (after 5000 stimulations), another 15 neurons were recruited to the assembly and, finally, at about 420,000 (after 7000 stimulations) another 5 neurons were further added to the assembly.

Consistently with the above observations, we find that the assembly size increased as the simulation progressed (Fig 5B). After the formation of the assembly, shown in the inset of the panel, we observe that before about 20,000 a.u. there was hardly any change in the assembly size, but then an increased number of non-stimulated units got recruited into the assembly. More specifically, there was a monotonic increase of the assembly size until about 200,000 a.u., where the assembly size started to increase more slowly, finally reaching a size of 45 neurons at 420,000 a.u. This slowing down of the increase of the assembly size was determined by the balance between the assembly increase regulated by the learning process, the decrease given by the forgetting mechanism and the input fields' normalization factors, which made it less likely for non-stimulated neurons to join the assembly for relatively large assembly sizes (see S1 Fig for observing the network evolution for a 10 times longer simulation).

Initially, the mean connection strength between the non-stimulated and stimulated neurons was relatively weak (Fig 5C, each curve represents the mean connection of a non-stimulated neuron with all 10 stimulated neurons). Then, as soon as the connection of a non-stimulated neuron with the stimulated assembly got strong enough to sustain its coactivation with the stimulated neurons, that connection was further strengthened, until reaching the maximum synaptic strength ($w_{max} = 0.3$).

## Assembly evolution for different stimulation frequencies

Our work draws inspiration from experimental findings in the human MTL suggesting that stimuli presented more frequently are encoded by larger assemblies of neurons. In order to study the relationship between the assembly size and the stimulation frequency within our model, we first considered the case of a network learning a single pattern presented at different frequencies.

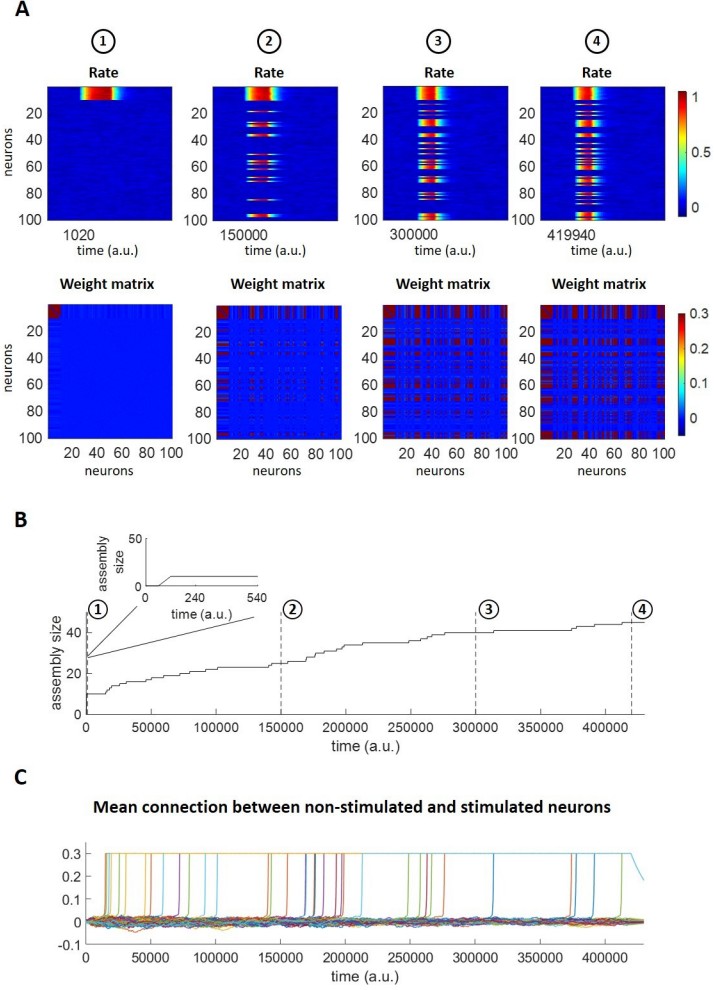

**Fig 5. Assembly evolution upon stimulus repetition.** A) Firing rates (top) and connections (bottom) of all 100 neurons at different stages of the assembly evolution. The number of active neurons increases over time. B) Number of assembly neurons over time. Initially only the 10 stimulated neurons respond. Inset: zoom with enlarged time scale. C) Mean connections of non-stimulated neurons with stimulated neurons. For each of the non-stimulated neurons, the average of its connections with the 10 stimulated neurons is shown. If a neuron joins the assembly, its mean weight value goes to $w_{max} = 0.3$.

Stimulations were given with frequencies $f = \frac{1}{30 \ a.u.}$, $f = \frac{1}{40 \ a.u.}$, $f = \frac{1}{60 \ a.u.}$ $f = \frac{1}{120 \ a.u.}$. As expected from our working hypothesis, there was a larger increase of the assemblies that were more frequently stimulated (Fig 6A). To further study this effect, in Fig 6B we show the assembly sizes at four stages of the simulations for the four frequencies tested. At stage 1, shortly after the initial assembly formation, all assemblies had the same size, but in the following 3 stages there was a significant difference between all the frequencies (in all cases, an ANOVA comparing all frequencies gave p $< 10^{-12}$ and the post-hoc t-test comparisons gave p $< 10^{-7}$).

Looking at the neurons' responses, at the four stages defined above and for the four different frequencies (see Fig 6C for one exemplary simulation), we observe that for higher stimulation frequencies more neurons were recruited into the assembly, in line with the previous plots.

A question arises of whether the increase in assembly size that we observe is dependent on the stimulation frequency or on the number of stimulations. Therefore, in the left panel of

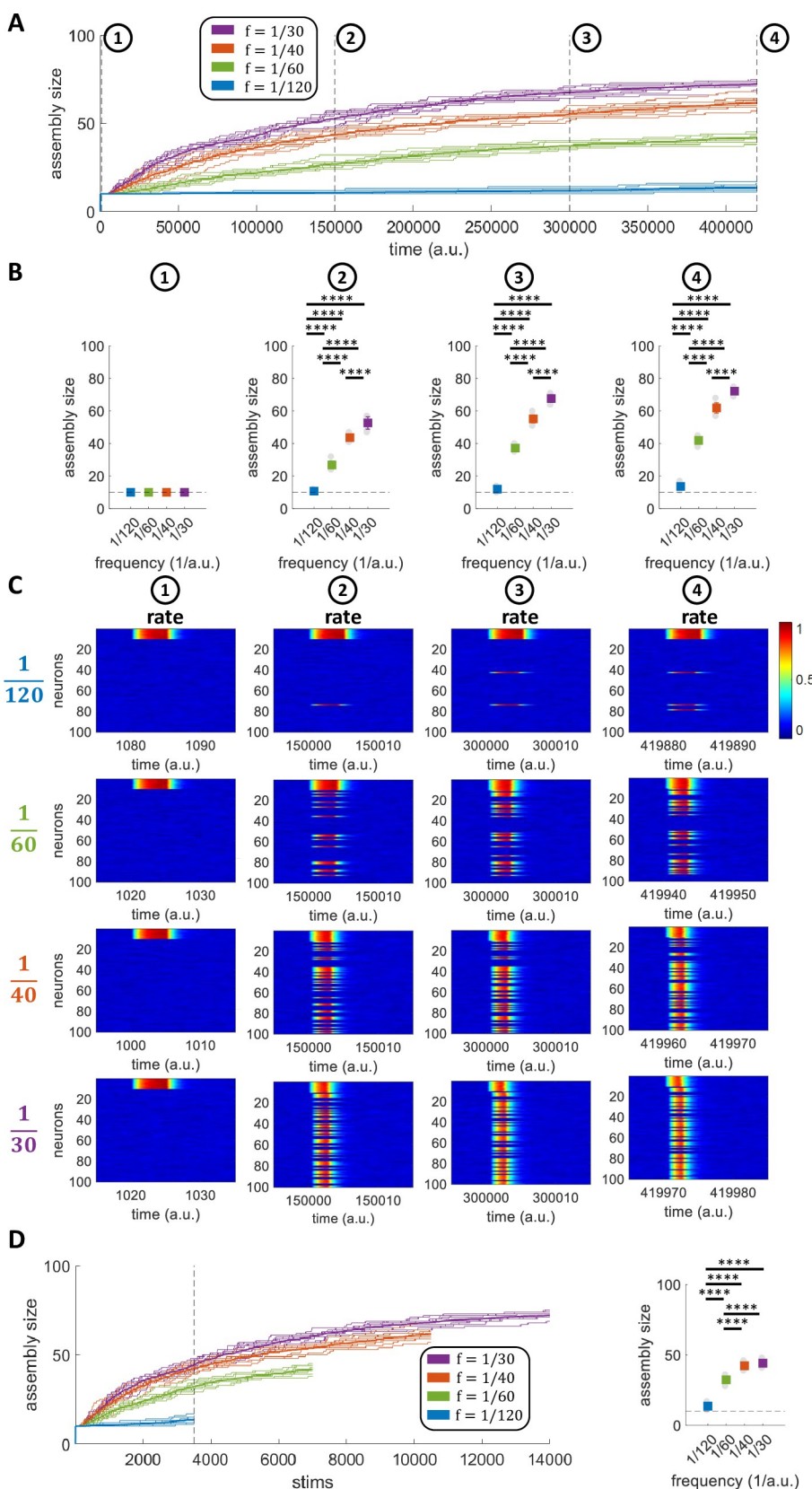

**Fig 6. Assembly evolution for different stimulation frequencies.** A) Number of assembly neurons over time for networks stimulated with 4 different frequencies. The bold lines indicate the mean over all the repetitions ($n = 10$) while the background lines represent the single repetitions. B) Assembly size for each of the frequencies at different times. The coloured squares indicate the mean+std while the grey circles in the background represent the single repetitions' results. C) Firing rates for all neurons at different times. For each frequency, an exemplar simulation is considered. D) Number of assembly neurons over stimulation number. The bold lines indicate the mean over all the repetitions ($n = 10$) while the background lines represent the single repetitions. On the right, the assembly size reached after 3500 stimulations for each of the frequencies is shown.

Fig 6D, we plot the assembly sizes for the 4 frequencies tested as a function of the number of stimulations, while the right panel shows the assembly sizes after 3500 stimulations (marked with a dotted vertical line in the left panel). As before, there was a significant difference of the assembly sizes for the different frequencies (ANOVA, $p < 10^{-12}$), and the post-hoc comparisons showed that these differences were significant in all cases (t-test, $p < 10^{-12}$), except between $f = \frac{1}{30 \, a.u.}$ and $= \frac{1}{40 \, a.u.}$. The fact that the assembly evolution was linked to the stimulation frequency, rather than solely to the number of stimulations, is due to the contribution of the forgetting term (Eq (8)), which had a stronger effect for the lower frequencies. This is because the forgetting term had more time to bring the increase of weight produced by the Hebbian learning back to zero, while having a less prominent effect for the higher frequencies and thus the similar assembly sizes after 3500 stimulations.

## Assembly evolution with two concurrent patterns

Having assessed the relationship between stimulation frequency and assembly growth in a single-memory network, we then examined the case of a network storing and retrieving two non-overlapping patterns. Here we aimed to investigate if and how the evolution of each assembly might influence the other one (e.g. the possibility that the assemblies take neurons from each other, or that they recruit the same neurons that were initially not part of any assembly). We were particularly interested in investigating if initially orthogonal neural assemblies would remain orthogonal when they evolve, or if, alternatively, one of the assemblies would show a progressively larger increase and eventually recruit neurons from the other one, showing a 'winner takes all' dynamics. First, we stimulated the network with two rectangular pulse trains, given at different times and at the same frequency ($f_1 = f_2 = \frac{1}{60 \, a.u.}$) to two non-overlapping assemblies. We found that the average size (over 10 simulations) of the two assemblies increased similarly over the simulation (Fig 7A). It can also be noted that in none of the simulations the difference between the assemblies' sizes increased over time, showing that there was not a 'winner takes all' dynamics. To further quantify the evolution of the assemblies, we considered three different stages (t = 1000 a.u.; t = 200000 a.u. and t = 420000 a.u.), from the formation of the assemblies to their reinforcement.

We examined the firing rates of all neurons and the connectivity between them in order to establish which neurons were recruited by the assemblies at each stage (see Fig 7B for one exemplary simulation). In the first stage, we observe the two formed assemblies (P1 and P2)–the first involving neurons 1 to 10 and the second one involving neurons 11 to 20 –each of them stimulated at different times and (still) not recruiting other neurons. The weight matrix shows two well separated clusters without connections between them or with the other neurons. At stage 2, we observe that each pattern had recruited neurons that initially were not part of the two assemblies (i.e. neurons 21 to 100). This increase in the number of neurons of each assembly is further observed at stage 3. Notably, from a close observation at the firing rate matrices we observe that P1 and P2 recruited different neurons and no neuron is shared by the two patterns, indicating that patterns remain orthogonal. In line with this observation, the

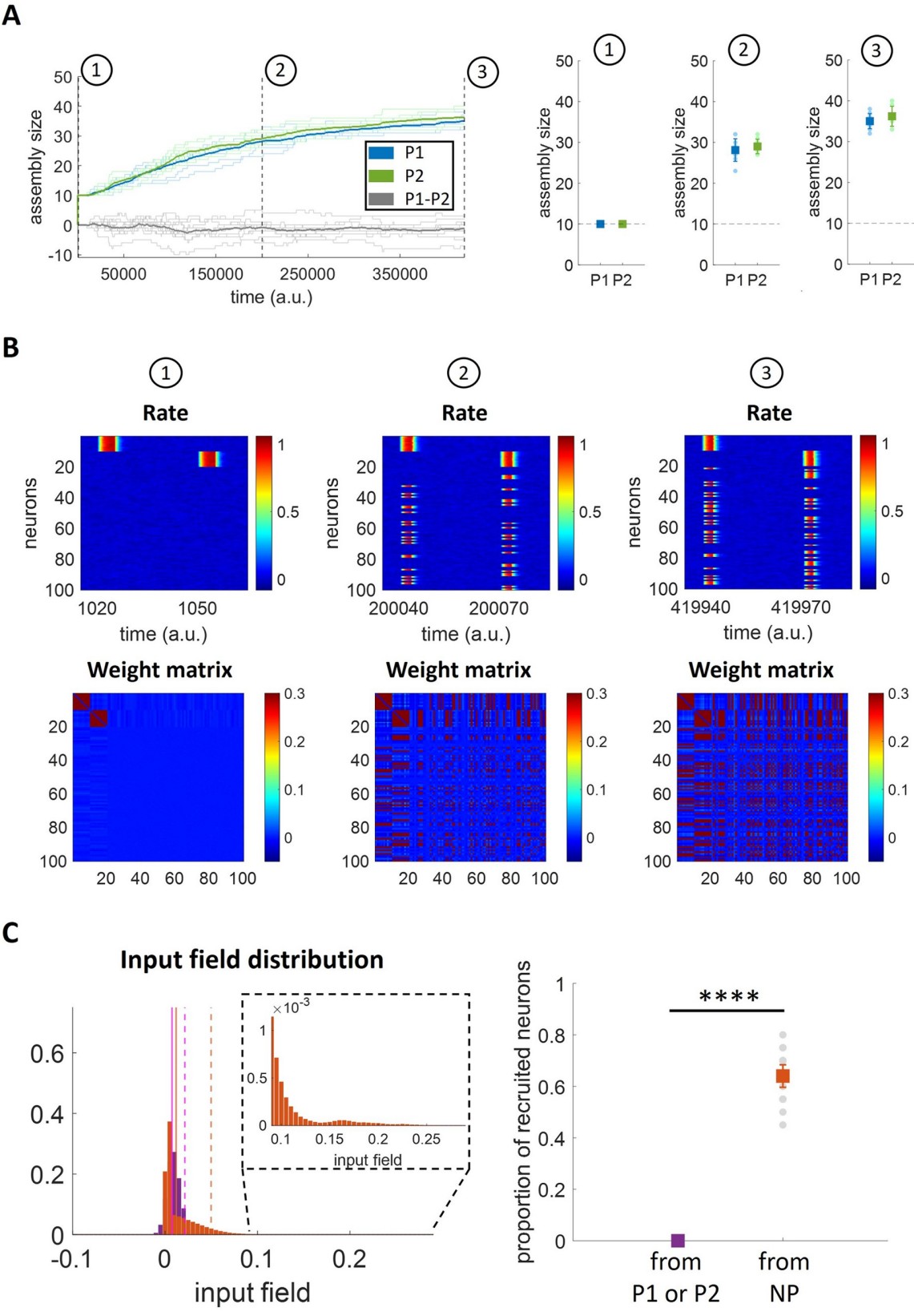

**Fig 7. Assembly evolution with two concurrent patterns stimulated at the same frequency.** A) Number of neurons per assembly over time. Left: The bold lines indicate the mean assembly sizes over all the repetitions ($n = 10$) while the background lines represent the single repetitions. The background grey lines indicate the difference of the assemblies' sizes for each repetition while the bold grey line represents the average difference. Right: The coloured squares indicate the mean+std while the circles in the background represent the single repetitions' results. B) Firing rates for all neurons and weight matrices at different times for a single simulation. C) Quantification of the higher tendency of recruiting neurons with weak synaptic connections. Left: Input field distributions for: P1 (neurons 1 to 10) while stimulating P2 (neurons 11 to 20), in purple; P2 while stimulating P1, in purple; other neurons (neurons 21 to 100; in case recruited, trials after recruitment are excluded) while stimulating P1 or P2, in orange. For each of the two distributions, the vertical solid line represents the mean while the dashed line corresponds to the 95th percentile; inset: zoom with enlarged y-axis scale. Right: Proportion of recruited neurons from P1 (neurons 1 to 10), P2 (neurons 11 to 20), NP (Non-Pattern, neurons 21 to 100). For both panels, all 10 simulations are included.

weight matrices show no cross-connections between the two assemblies. The result of maintaining orthogonality between patterns is a consequence of the synaptic normalization—implemented in the model through the factor $S_{w,i}(t)$—which normalizes the input field of each neuron according to the summed connection weight. The normalization factor $S_{w,i}(t)$ downscales the input fields of the neurons belonging to an assembly, thus making them less likely to be recruited by the other assembly compared to the other (not yet recruited) neurons.

In order to illustrate how $S_{w,i}(t)$ downscales the input fields of assembly neurons, the left panel of Fig 7C depicts the distribution of the input fields (i.e. the variable h defined in Eq 2) of the stimulated neurons of P1 and P2 (purple) when stimulating the other assembly (i.e. the input field of P1 when stimulating P2 and vice versa), as well as the input fields of the neurons not belonging to these assemblies (orange), excluding, for any unit later recruited into either P1 or P2, the stimulation trials after which it had been recruited. We observe that the input field distribution of the neurons not part of either assembly reaches higher input field values, thus being more likely for those neurons to fire together with the stimulated neurons and join the assembly. As a consequence of this, only neurons not belonging to any assembly are available to be recruited, thus making the difference between recruiting neurons from one of the two stimulated assemblies (i.e. neurons 1 to 20) and those initially not belonging to an assembly (i.e. neurons 21 to 100) highly significant (t-test; $p < 10^{-12}$) (Fig 7C, right).

Next, we repeated the simulations with two non-overlapping neuronal assemblies, but now using different frequencies of stimulation ($f_1 = \frac{1}{60\ a.u.}$; $f_2 = \frac{1}{120\ a.u.}$). As expected, we observe that the two patterns had a different evolution over time (Fig 8A) (see S6 Fig for direct comparison with the evolution over time observed in Figs 6A and 7A). Starting with an equal assembly size immediately after their formation (stage 1), the pattern that was stimulated less frequently barely increased its size, whereas the other one showed a marked increase with a size that was significantly larger than the one of the other pattern in stages 2 and 3 marked in the plot (t-test, $p < 10^{-12}$ in both cases). As for the previous simulations, the analysis of the firing of all the neurons in the network and their connectivity pattern allows to assess which neurons were recruited by the two assemblies at the three stages of the simulations (see Fig 8B for one exemplary simulation). In the first stage, we observe the two distinct patterns (the one on top, from neuron 1 to 10, is the one stimulated more frequently), and, as the simulation progressed, in stages 2 and 3 we notice that the pattern stimulated with a high frequency (and not the other one) recruited other neurons, but none of these corresponded to the other pattern. As before, this was due to the fact that the other neurons (i.e. those not belonging to a pattern) reached higher values of the input field (left panel of Fig 8C) and were therefore recruited in a significantly higher proportion (right panel of Fig 8C) (t-test; $p < 10^{-12}$).

Summarizing, in none of our simulations (both using patterns stimulated with the same frequency or with different frequencies) we obtained an overlap between the patterns, which remained orthogonal after changing their sizes. Patterns stimulated with non-overlapping

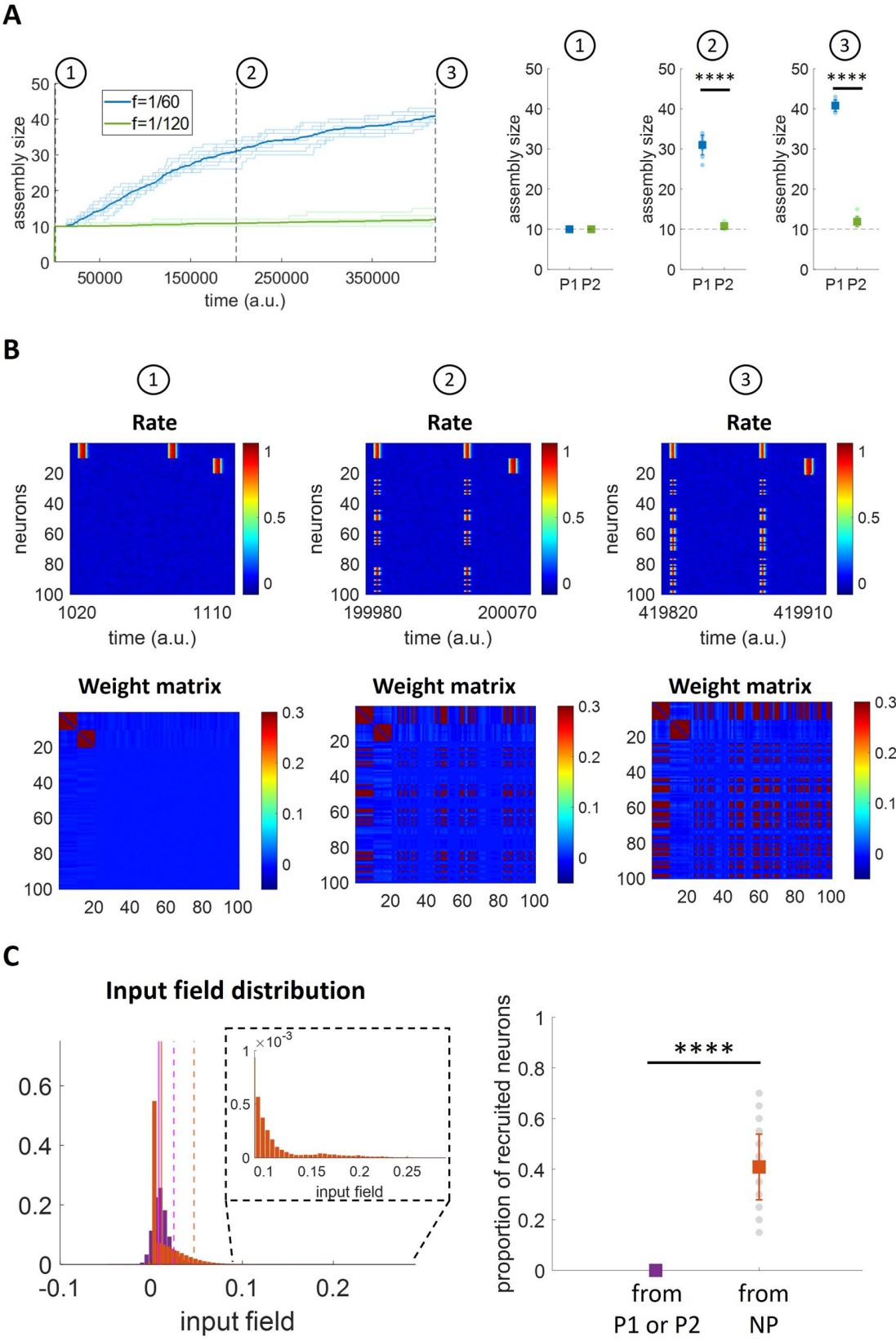

**Fig 8. Assembly evolution with two concurrent patterns stimulated at different frequencies.** A) Number of neurons per assembly over time. Left: The bold lines indicate the mean assembly sizes over all the repetitions (n = 10) while the background lines represent the single repetitions. Right: The coloured squares indicate the mean+std while the circles in the background represent the single repetitions' results. B) Firing rates for all neurons and weight matrices at different times for a single simulation. C) Quantification of the higher tendency of recruiting neurons with weak synaptic connections. Left: Input field distributions for: P1 (neurons 1 to 10) while stimulating P2 (neurons 11 to 20), in purple; P2 while stimulating P1, in purple; other neurons (neurons 21 to 100; in case recruited, trials after recruitment are excluded) while stimulating P1 or P2, in orange. For each of the two distributions, the vertical solid line represents the mean while the dashed line corresponds to the 95th percentile; inset: zoom with enlarged y-axis scale. Right: Proportion of recruited neurons from P1 (neurons 1 to 10), P2 (neurons 11 to 20), NP (Non-Pattern, neurons 21 to 100). For both panels, all 10 simulations are included.

stimuli remained orthogonal even when they took all the neurons of the network, which happened in the case of much longer simulations (see S3 Fig).

## Assembly reinforcement and forgetting

Next, we directly tested the hypothesis illustrated in Fig 1B, namely that, after the formation of 3 similar assemblies, the one stimulated more frequently will increase its size, the one stimulated less frequently will show a smaller increase, whereas the third one, not further stimulated after its formation, will disappear. For this, in the first phase (assembly formation), 3 non-overlapping patterns (P1, P2 and P3; involving neurons 1 to 10, 11 to 20 and 21 to 30, respectively) were stimulated through rectangular pulse trains given at different times and at the same frequency ($f_1 = f_2 = f_3 = \frac{1}{90 \ a.u.}$). In the second phase (assembly evolution), P1 and P2 were stimulated at different frequencies ($f_1 = \frac{1}{60 \ a.u.}$, $f_2 = \frac{1}{120 \ a.u.}$) and P3 was not further stimulated ($f_3 = 0$).

After the first phase, 3 distinct assemblies of equal size were formed (insets in Fig 9A; see also the firing response and the connectivity matrix in Fig 9C and 9D at stage 1, immediately after the formation of the assemblies). Then, during the second stimulation phase, the 3 assemblies reached different sizes (Fig 9A, top; see also Fig 9C and 9D at stages 2, 3 and 4). The mean value of the synaptic weights inside P1 and P2 remained stable after the initial assembly formation since they were repeatedly stimulated, while the ones between the neurons of P3 decreased exponentially, due to the effect of the forgetting term (Fig 9A, bottom).

It is of interest to look at the mean connections of each of the non-stimulated neurons (i.e. neurons 31 to 100) and those of P2 (i.e. neurons 11 to 20) and P3 (i.e. neurons 21 to 30) with P1 (i.e. neurons 1 to 10) in order to study which neurons were recruited by P1 and at which stage of the simulations (Fig 9B, all simulations were considered). In the first stages of the simulations, non-stimulated neurons were recruited by P1, but not the neurons of P2 and P3. As the simulations progressed, when the weight between the neurons of P3 decayed close to baseline levels, P3 disappeared and its neurons started to be recruited by P1. In other words, once the memory was forgotten, the neurons representing it became available to join the representation of other memories. Since P2 continued to be stimulated, the average weight between its neurons continued to be at the same level and, consequently, none of the neurons of P2 were recruited by the more frequently reinforced P1.

As for the previous simulations, we looked at the firing of the neurons and the connectivity matrix at subsequent stages of the simulation (see Fig 9C for one exemplary simulation). At stage 1, we observe the 3 non-overlapping patterns shortly after their formation. At stage 2, P3 was no longer stimulated and the corresponding clustering in the weight matrix disappeared; besides, we observe that one of the neurons initially belonging to P3 (i.e. neuron 29) already responded when stimulating P1. At the later stages (stage 3 and 4) we observe that P1 had recruited more neurons of P3 and of the non-stimulated neurons, but not of P2. Note also that in the connectivity matrix there was not an increase in the weights between P1 and P2,

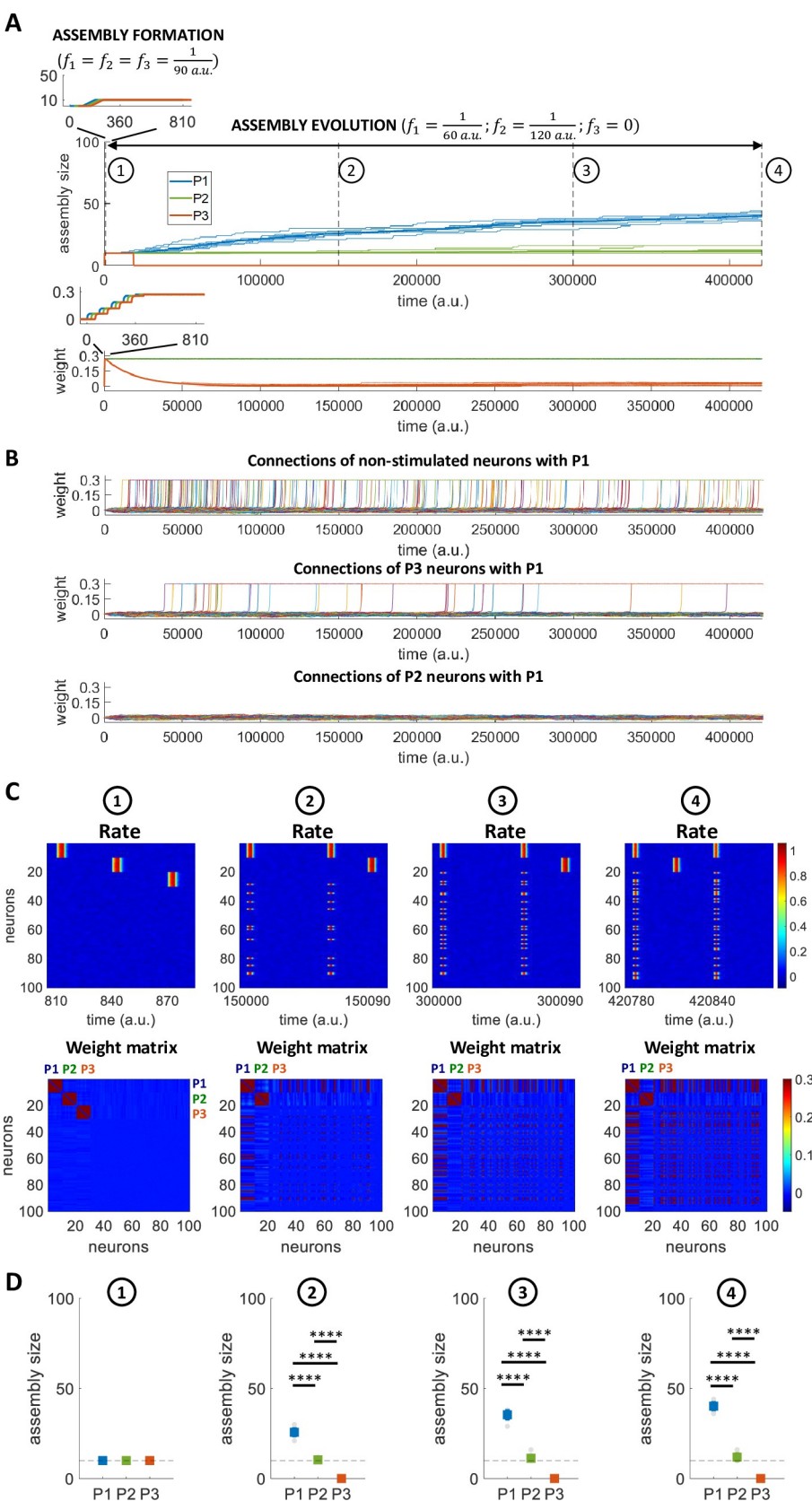

**Fig 9. Assembly reinforcement and forgetting.** A) Number of neurons per assembly and mean synaptic weight over time. The bold lines indicate the mean over all the repetitions ($n = 10$) while the background lines represent the single repetitions. B) Mean connections with P1 (neurons 1 to 10) (All simulations). Top: for each of the non-stimulated neurons (neurons 31 to 100), the average of its connections with P1 is shown. Middle: for each of the P3 neurons (neurons 21 to 30), the average of its connections with P1 is displayed. Bottom: for each of the P2 neurons (neurons 11 to 20), the average of its connections with P1 is shown. C) Firing rates and weight matrices for all neurons at different times for a single simulation. The times 1 to 4 correspond to the times considered in the above panel. D) Number of neurons per assembly at different times. The coloured squares indicate the mean+std while the grey circles in the background show the single repetitions' results.

meaning that the two patterns remained separated. In line with these results, we find equal sizes for the 3 assemblies shortly after their formation (stage 1), but different sizes between them as the simulations progressed (ANOVA, $p < 10^{-12}$; post-hoc t-tests, $p < 10^{-11}$ in all cases), with P1 increasing its size, P2 remaining approximately with the same size and P3 disappearing (Fig 9D).

## Scalability of the model

To speed up calculations, in the previous simulations we have used a network of N = 100 and patterns initially having 10 neurons each. Next, we tested the robustness of our model to changes in the network size ($N$) and the fraction of stimulated neurons ($N \cdot \gamma$). In particular, we further evaluated the network dynamics during the formation and evolution of 1 assembly for 2 different stimulation frequencies ($f = \frac{1}{30\ a.u.}$ and $f = \frac{1}{60\ a.u.}$) and other combinations of $N$ and $N \cdot \gamma$ (N = 500 and $N \cdot \gamma$ = 10; N = 500 and $N \cdot \gamma$ = 50; N = 1000 and $N \cdot \gamma$ = 10; N = 1000 and $N \cdot \gamma$ = 50).

As before (see Fig 6A), for both frequencies we observe an increase of the assembly sizes with time, which was higher for the simulations with the higher frequency of stimulation (Fig 10). We examined the assembly increase at three different stages of the simulations (Fig 10A, bottom; Fig 10B, bottom). For both frequencies and both network sizes the increase was proportionally larger for the sparser pattern, as in this case there were more neurons that could be recruited into the pattern. There was, however, a clear difference compared to the simulation with 100 neurons in Fig 6, which is due to the fact that in case of larger networks the assemblies reached a relatively stable regime faster (in less than ~200,000 $a.u.$ for N = 500, and in less than ~100,000 $a.u.$, for N = 1000, in Fig 10 compared to about less than ~400,000 $a.u.$ in Fig 6). This is because increasing the number of non-stimulated neurons (i.e. $N-N \cdot \gamma$) gives a higher probability of recruiting neurons, which results in a faster assembly evolution for our simulations with N = 500 and N = 1000 (and in a faster assembly evolution for our simulations with N = 1000 compared to N = 500). Nevertheless, at the plateau level the final relative assembly sizes with N = 100, N = 500 and N = 1000 are comparable since the effect of the forgetting term (β) and the normalization factors ($S_{W,i}(t)$ and $S_{R,i}(t)$) does not change.

We showed the robustness of the model to changes in network size (comparing two further network sizes N = 500 and N = 1000 to the standard network size N = 100) and to changes in assembly size (testing both $N \cdot \gamma$ = 10 and $N \cdot \gamma$ = 50 for each of the new network sizes) (Fig 10). Furthermore, we decided to test the robustness of our model to changes in the number of stimulated assemblies. In particular, we evaluated the network dynamics within a network of N = 1000 using a different combination of number of stimulated assemblies and $N \cdot \gamma$ (2 assemblies of $N \cdot \gamma$ = 10; 2 assemblies of $N \cdot \gamma$ = 20; 10 assemblies of $N \cdot \gamma$ = 10; 10 assemblies of $N \cdot \gamma$ = 20; 20 assemblies of $N \cdot \gamma$ = 10; 20 assemblies of $N \cdot \gamma$ = 20;), for 2 different stimulation frequencies ($f_1 = \frac{1}{600\ a.u.}$ and $f_2 = \frac{1}{1200\ a.u.}$). In each simulation, half of the assemblies were stimulated with $f_1 = \frac{1}{600\ a.u.}$ and the other half with $f_2 = \frac{1}{1200\ a.u.}$. It can be noticed that the frequencies adopted

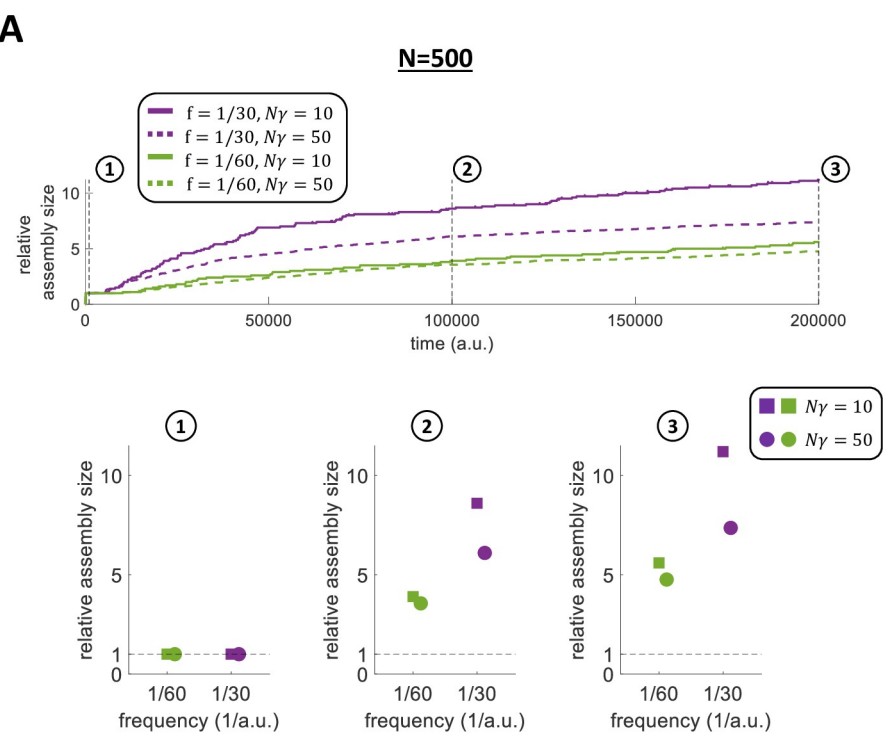

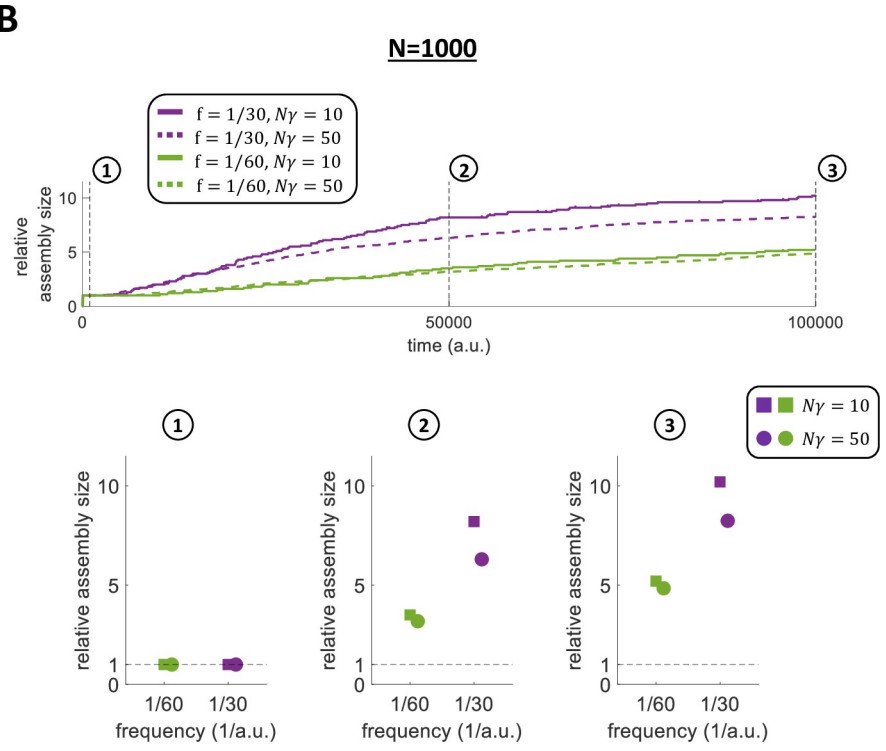

**Fig 10. Assembly evolution with larger networks.** A) Relative assembly size over time within a network of 500 neurons for different numbers of stimulated neurons ($N = 500$ and $N \cdot \gamma = 10$; $N = 500$ and $N \cdot \gamma = 50$), and for 2 different stimulation frequencies. The relative size (RS) at time m is calculated as $RS_m = \frac{SIZE_m}{N \cdot \gamma}$. Bottom: relative assembly size at different times for each of the 2 frequencies and each of the 2 stimulated pattern sizes. B) Relative assembly size over time within a network of 1000 neurons for different numbers of stimulated neurons ($N = 1000$ and $N \cdot \gamma = 10$; $N = 1000$

and $N \cdot \gamma = 50$), and for 2 different stimulation frequencies. The relative size (RS) at time m is calculated as $RS_m = \frac{SIZE_m}{N \cdot \gamma}$. Bottom: relative assembly size at different times for each of the 2 frequencies and each of the 2 stimulated pattern sizes.

here ($f_1 = \frac{1}{600\ a.u.}$ and $f_2 = \frac{1}{1200\ a.u.}$) were lower than the frequencies used throughout the rest of the work ($f = \frac{1}{30\ a.u.}$; $f = \frac{1}{40\ a.u.}$; $f = \frac{1}{60\ a.u.}$; $f = \frac{1}{90\ a.u.}$; $f = \frac{1}{120\ a.u.}$;). The reason is that, when stimulating a larger number of assemblies at different times, the stimulation frequencies need to be lower in order to avoid temporal overlaps between consecutive simulations of different assemblies. Consequently, the forgetting term needs to be adjusted, as discussed in Methods.

In all the simulations with N = 1000 and with different combinations of number of assemblies and $N \cdot \gamma$, we found qualitatively similar results compared to the findings previously shown (Figs 11 and S4). Specifically, we again found that: i) assemblies stimulated with the same frequency increase their sizes similarly, while the sizes of assemblies stimulated with different frequencies evolve significantly differently (Fig 11; t-test); ii) the stimulated assemblies change their sizes recruiting different neurons, thus there was no overlap of neurons between different assemblies (i.e. the different assemblies remained orthogonal) (S4 Fig).

## Discussion

Our memories are constantly changing. They are created, consolidated into long-term representations and sometimes forgotten. In contrast to these dynamical memory processes, standard attractor neural networks are static, in the sense that memories are stored as specific patterns of network activations that remain the same [22]. Furthermore, in attractor neural networks, memories correspond to fixed points and, once a memory is reached, the network stays there forever, whereas in real life we tend to go from one memory to another. To tackle these issues, we here proposed a "dynamic memory model" by implementing a biologically-plausible online Hebbian learning rule in a rate attractor neural network model. To avoid that the network remains permanently in a fixed point, we also added an adaptation mechanism, as in previous works [38,41–44], with which, after the offset of the stimulus presentation, the network went down to baseline.

Rate attractor models with online learning have previously been studied in order to investigate several aspects of the learning dynamics [26–32]. Specifically, earlier studies [26–30] aimed at characterizing the network storage capacity when distinct patterns are stored at subsequent times. More recent studies [31,32] have proposed a method for inferring learning rules from in vivo data and implementing those rules in firing rate models generating attractor dynamics. However, in all those studies the evolution of memory representations with stimulus repetitions was not investigated. In contrast to these works, our primary goal was to replicate the dynamics of memory processes (formation, reinforcement and forgetting). For this, we have: i) used an explicit learning rule acting continuously on the network connectivity; ii) examined the effects of presenting patterns at different frequencies; iii) considered the role of background activity in shaping long-term memory representations.

Since recurrent networks with synapses regulated by Hebbian learning become unstable [47,48], several works using attractor neural networks with spiking neurons have implemented compensatory mechanisms [35–37], based on heterosynaptic plasticity occurring on synapses connecting neurons that are not activated, in parallel to Hebbian plasticity (which happens instead on synapses connecting co-activated neurons) [49–51]. In our work, we further limited the runaway dynamics of Hebbian learning through two mechanisms of normalization of the input. One mechanism (synaptic normalization) decreased the synaptic efficacy when the postsynaptic neurons built strong positive connections. The other mechanism (divisive

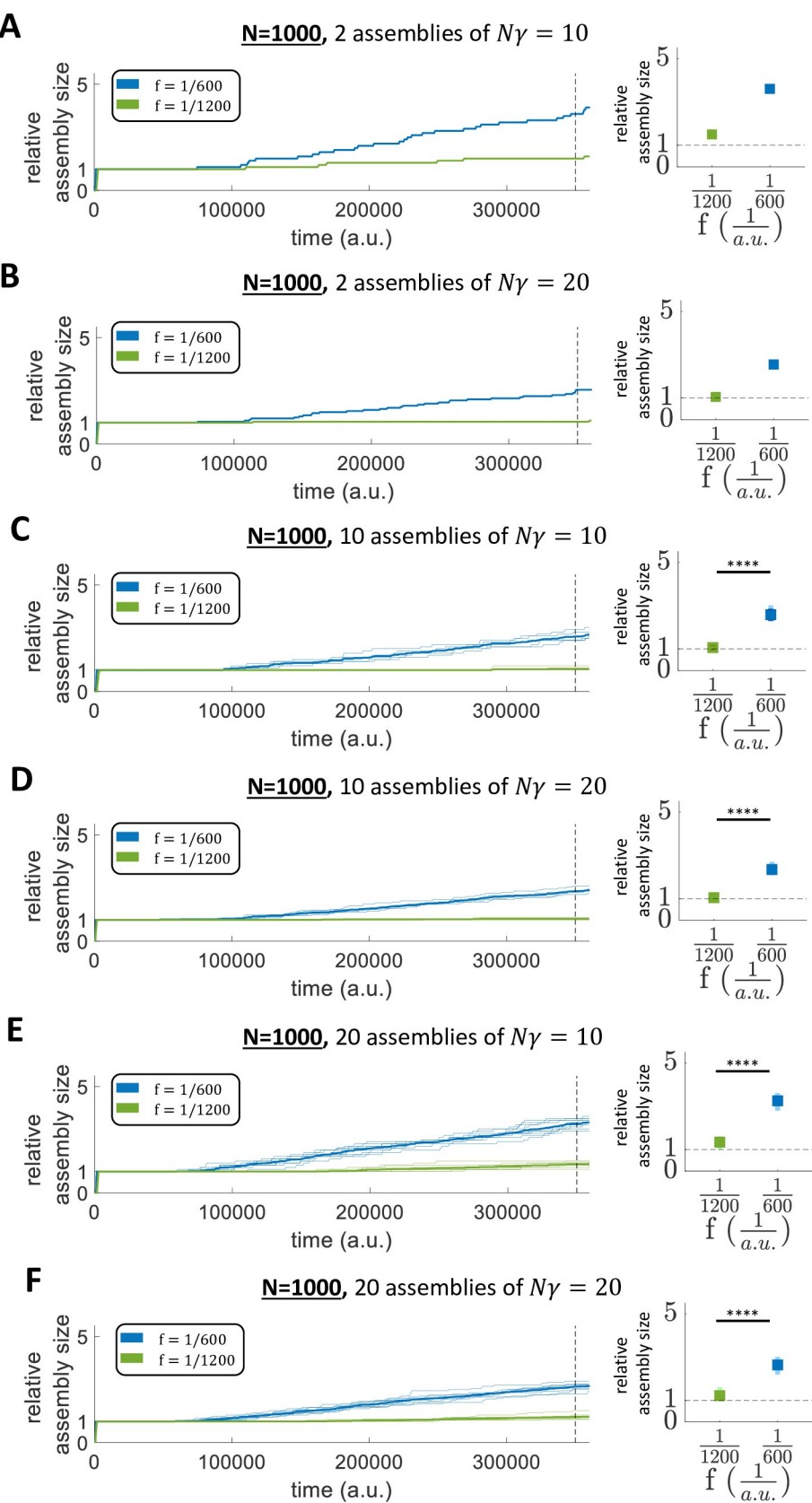

**Fig 11. Assembly evolution for different stimulation frequencies in larger networks with different number of assemblies and number of stimulated neurons.** A-B-C-D-E-F) Relative assembly size over time within a network of 1000 neurons for: A) 2 assemblies of 10 stimulated neurons ($N \cdot \gamma = 10$); B) 2 assemblies of 20 stimulated neurons; C) 10 assemblies of 10 stimulated neurons; D) 10 assemblies of 20 stimulated neurons; E) 20 assemblies of 10 stimulated neurons; F) 20 assemblies of 20 stimulated neurons. The bold lines indicate the mean over all the assemblies stimulated at the same frequency while the background lines represent the single assembly sizes. Right: Assembly size for each of the two frequencies at $t = 350000$ *a.u.* The bold squares indicate the mean+std across the assemblies stimulated at the same frequency while the squares in the background represent the single assemblies' results. (Parameters: $\beta = 0.000125$.).

normalization) modulated the input received by the postsynaptic neurons according to the mean activity of the network. In our model these two compensatory processes exerted two distinct functional roles in memory coding. The divisive normalization promoted sparseness of the neural coding (i.e. the information tended to be represented by a limited subset of neurons of the network), in line with the high sparseness of human MTL neurons [52,53]. Then, the synaptic normalization prevented neurons that were already part of an assembly from responding by chance to other stimuli, thus avoiding interference between the assemblies.

Our main working hypothesis was that the size of an assembly representing a memory dynamically changed according to the frequency of presentation of the pattern corresponding to that memory, due to the interplay of Hebbian learning and background firing activity. In line with this, we showed that the chance of reinforcing the connections between neurons of an assembly and other neurons significantly increased with the frequency of stimulation of the corresponding memory pattern–i.e. the higher the stimulation frequency, the higher the assembly size reached at the end of the stimulation. In particular, we found that the repeated presentation of a pattern lead, first, to the formation of an assembly representing it and, then, to its reinforcement, by increasing the number of assembly neurons, if the frequency of presentation was large enough, or its forgetting, if the frequency was too low. This way, we could study how the size of a memory representation was linked to the familiarity of the stimulus, in line with behavioural results showing a clear link between memory performance and familiarity [18–20], fMRI results showing that the presentation of personally relevant visual stimuli generates stronger hippocampal activation than less relevant stimuli [17], and results obtained with direct single neuron recordings in the human hippocampus showing that more neurons tend to represent familiar, personally relevant items [16]. In line with these results one could expect that the repeated presentation of novel stimuli (the face of an initially unknown person) would create an increasingly larger neuronal representation, which, due to the very sparse sampling of human hippocampal neurons allowed by current recordings, would give an increasing tendency to find single neuron responses, the more the stimulus is presented. Our results show the role of synaptic normalization in reducing the firing of the neurons that are already recruited into an assembly, thus decreasing the probability to be recruited by other assemblies and keeping assemblies orthogonal. Based on this result, we can propose two experimental predictions: i) neurons with low firing rates are already part of consolidated memory assemblies and are therefore less likely to respond to novel stimuli (i.e. novel stimuli will tend to recruit neurons with relatively high firing rates); ii) the neurons that start firing to novel stimuli should decrease their baseline firing rates with the repeated presentation of the stimuli. It should also be noted that in our simulations the reinforcement of the assemblies was obtained by the repeated presentation of the corresponding patterns. However, it is conceivable that such reinforcement can not only be created by external stimulation but also by internal processes, given by a stochastic reactivation of the memory patterns [54,55,56], a process that is particularly prominent during sleep [57,58].

We also showed that, when multiple non-overlapping assemblies were regularly stimulated, they did not interfere with each other. In fact, if stimulated at the same rate, none of the

assemblies outweighed the other and they evolved independently–i.e. without taking neurons from each other and recruiting neurons not belonging to any assembly. In other words, the presence of two or more patterns in a network did not lead to a 'winner takes all' dynamics, in which one of the assemblies outweighed the other one, or overlapping representations, with neurons being part of more than one pattern, thus generating sparse representations, as found in human MTL recordings [52,53].

Memory relies on the coding of associations—for example, to remember meeting a particular person in a particular place—a process that involves the hippocampus [15,59,60]. Based on results obtained with single cell recordings in the human hippocampus, namely, that if a neuron responds to two or more concepts, these concepts tend to be associated [61–63], we suggested that in this area associations are encoded via partial overlaps between the assemblies representing the concepts involved [15,53]. In a previous modelling study [38], we have shown that such partial overlaps are efficient to encode and retrieve associations. This model assumes neglectable amounts of overlaps between non-associated items and it is therefore important that non-associated items (as the ones in this study) form non-overlapping representations, or otherwise, overlaps that may have been created by chance would be erroneously interpreted as encoding meaningful associations. Future studies may show how overlaps between assemblies representing associations might be created by showing patterns together, but not too often, or otherwise the patterns will have a large overlap that will merge them into a single fixed point [38]. We should also mention that an alternative (and not mutually exclusive) model for enlarging familiar assemblies is that, via partial overlaps, the assembly representing a concept that is experienced in different contexts may incorporate neurons representing these contexts, given that the context where a concept is experienced (i.e. a particular place) is another concept with a corresponding assembly.

Finally, we have implemented a forgetting mechanism as an ongoing weakening of the synapses' strength which continuously competes with the synapses' potentiation [2,8,45,46]. Due to this forgetting mechanism, assemblies representing memories that are no longer presented end up disappearing. Moreover, we showed that, once an assembly disappeared due to lack of stimulation, its neurons became available to be recruited by other (further stimulated) memories, becoming part of their corresponding assemblies. This increases the coding flexibility of the system, forgetting old memories that are no longer relevant, in favour of others that are revisited more frequently and become more relevant.

## Methods

### Standard attractor model

The starting point for the construction of the dynamic attractor model was a standard attractor neural network with static connections within the network, pre-defined according to the memories to be encoded and stored as unipolar binary patterns ($\xi_i^\chi \in \{0, 1\}$, for all neurons $i$).

The model was built as an attractor neural network of N rate neurons whose dynamics is described by:

$$\tau_r \frac{dr_i}{dt} = -r_i(t) + r_0 + \phi(h_i(t)) \tag{1}$$

where $r_i(t)$ is the firing rate of a neuron $i$, $\tau_r$ is the neural activation time-constant, $r_0$ is the baseline firing rate, $\phi$ is a sigmoidal transfer function and $h_i(t)$ is the input field to the neuron $i$.

The input field is defined as:

$$h_i(t) = \sum_{i \neq j}^{N} w_{ij}(t) r_j(t) + I_i(t) \tag{2}$$

where $I_i(t)$ is the external input to the neuron $i$ and $w_{ij}(t)$ is the value of the connection going from neuron $j$ to neuron $i$.

The frequency-current (f-I) curve is a sigmoidal function:

$$\phi(h_i) = \frac{r_{max}}{1 + e^{-b(h_i - h_0)}} \tag{3}$$

where $h_0$ is the threshold parameter, $b$ the slope parameter and $r_{max}$ the maximal firing rate ($r_{max} = 1$).

## Dynamic attractor model

To the standard model described in the previous section we added: i) a neural adaptation mechanism, giving a decay of the activations after the removal of external stimulation (otherwise, the stimulated pattern would stay forever active); ii) Gaussian noise simulating ongoing background activity; iii) an online Hebbian learning rule, to dynamically update the synapses; iv) a relatively slow forgetting mechanism, to introduce an exponential decay of synaptic efficacies; v) compensatory mechanisms in order to avoid runaway dynamics.

## Neural adaptation

In attractor networks, if the network state $\vec{r}$ is driven to one stored memory $\chi$ (e.g. through the external input $\vec{I}$), all the neurons that participate in that memory $\chi$ will remain active after the removal of the external input $\vec{I}$. However, in a biologically plausible scenario, memories are continuously changing. Therefore, to avoid having the network being "stuck" in the attractor state after the stimulus offset, an adaptation mechanism [42] was implemented by defining, for each neuron $i$, an activity-dependent firing threshold $\theta_i(t)$ [41,43,44].

The dynamics of the firing threshold evolves as:

$$\tau_\theta \frac{d\theta_i}{dt} = -\theta_i(t) + \theta_0 + D_\theta r_i(t) \tag{4}$$

where $\theta_i(t)$ is the firing threshold of a neuron $i$, $\tau_\theta$ is the adaptation time-constant, $\theta_0$ is the base threshold in the absence of firing and $D_\theta$ is a constant that determines the strength of the adaptation.

The adaptive threshold $\theta(t)$ is inserted into Eq 3 as:

$$\phi(h_i, \theta_i) = \frac{r_{max}}{1 + e^{(-b[h_i(t) - \theta_i(t)])}} \tag{5}$$

where $\theta_i(t)$ is a moving threshold.

## Background activity

Part of the input arises from background activity which we assume to be normally distributed.

The dynamics of each neuron $i$ is defined as:

$$\tau_r \frac{dr_i}{dt} = -r_i(t) + r_0 + \phi(h_i(t), \theta_i(t)) + C \cdot \xi_i(t) \tag{6}$$

where $C$ is the coupling constant of the Gaussian noise, and $\xi_i(t)$ is the noise value sampled from a Gaussian distribution of mean $\mu = 0$ and standard deviation $\sigma = 1$. We note that, for simulation purposes, $C$ contains a factor $\sqrt{\frac{\Delta t}{\tau_r}}$ where $\Delta t$ is the timestep.

## Online Hebbian-like learning rule

We implemented a Hebbian-like learning rule [32], in which the dynamics of the synaptic connections is described by:

$$\tau_w \frac{dw_{ij}}{dt} = \eta(r_i - <r_i>)\left(r_j - <r_j>\right) \qquad (7)$$

where $w_{ij}$ is the value of the connection going from neuron $j$ to neuron $i$, the constant $\eta$ is the learning rate and $\tau_w$ is the learning time-constant. The term $<r_i>$ indicates the running average of the firing rate of neuron $i$ over a box-shaped time window of fixed length $T_{LR}$. The synaptic weights $w_{ij}$ have hard bounds at $[w_{min}, w_{max}]$, as in previous works [37,64,65,66].

## Forgetting mechanism

A learning rule as in Eq (7), during uncontrolled spontaneous activity, induces a random walk of the synaptic weights over time, in which the connection matrix has a stable average around 0, but its standard deviation keeps increasing. To stabilize the network, we introduced a "forgetting term" to the Hebbian-like learning rule, which is in line with forgetting mechanisms described in experimental psychology [45,46] and similar to synaptic decay implementations introduced in previous works [54],:

$$\tau_w \frac{dw_{ij}}{dt} = \eta(r_i - <r_i>)\left(r_j - <r_j>\right) - \beta w_{ij} \qquad (8)$$

where the constant $\beta$ regulates the timescale of forgetting.

## Compensatory mechanisms

**Synaptic normalization.** Hebbian plasticity, through long term potentiation (LTP), acts as a positive feedback mechanism: the potentiated synapses make the postsynaptic neurons more likely to fire and, consequently, those synapses are further strengthened. To counterbalance this effect, different forms of negative feedback mechanisms have been proposed and studied in the last decades [50,67]. In this regard, it has been proposed theoretically [68] and then shown experimentally with neurons of the primary visual cortex [51] that Hebbian potentiation of specific synapses is accompanied by weakening of their adjacent synapses by heterosynaptic plasticity.

In line with previous works [69], we introduced in our model a term that takes into account an input normalization depending on the summed weight of all strong connections. For each neuron $i$, this term is defined as:

$$S_{W,i}(t) = \frac{1}{\sqrt{1 + \alpha_w \sum_{i \neq j}^{N}(w_{ij}(t)H(w_{ij}(t) - w_{thr}))}} \qquad (9)$$

where $\alpha_w$ determines the strength of the scaling, $H$ is the Heaviside function and $w_{thr} = \frac{w_{max}}{6}$.

**Divisive normalization.** It has been proposed that a neuronal response is given by the ratio between the driving input from other neurons and a normalization signal proportional to the summed activity of the neuronal population [70,71]. This kind of normalization allows adjusting the neuron's dynamic range to changes in the input range [72], which is critical when the input range can vary substantially. In our work, since we plan to model the change of the sizes of neuronal assemblies, the neurons of these assemblies are subject to large changes in their recurrent inputs.

For this reason, we introduced in the model a term that takes into account an input normalization based on the mean network activity. For each neuron i, this term is defined as:

$$S_{R,i}(t) = \frac{1}{1 + \frac{\alpha_r}{\gamma N} \sum_{i \neq j}^{N} r_j(t)} \tag{10}$$

where $\alpha_r$ determines the strength of the normalization, N is the size of the network and $\gamma$ is the network sparseness defined as the fraction of directly stimulated neurons.

Introducing $S_{W,i}(t)$ and $S_{R,i}(t)$, the input field $h_i(t)$ for a neuron i is defined as:

$$h_i(t) = S_{R,i}(t)S_{W,i}(t)\left(\sum_{i \neq j}^{N} w_{ij}(t)r_j(t) + I_i(t)\right) \tag{11}$$

The choice of $\alpha_w$ and $\alpha_r$ (together with the choice of noise level, learning and forgetting rates and stimulation frequency) determines the maximum size that an assembly can reach (see S1, S2 and S3 Figs to observe which maximum size an assembly can reach in different conditions). Specifically, the more an assembly grows the more $S_{R,i}(t)$ prevents other neurons to join it, while $S_{W,i}(t)$ tends to contrast the assembly activation. Using a high value for $\alpha_r$ could produce a rebound activation of the stimulated assembly, while using a high value for $\alpha_w$ may lead the stimulated assembly to die out when reaching a certain size. In our work, we limited these possible effects of the normalization factors $S_{R,i}(t)$ and $S_{W,i}(t)$ while successfully limiting the runaway dynamics of Hebbian learning (see S1 Text for further considerations about this).

All the mechanisms used to stabilize the system (namely: hard bounds of the synaptic weights, weight decay, divisive normalization and synaptic normalization) were necessary for the functional behaviour of our model (see S5 Fig).

## Parameter choices

The model has several parameters, whose values are the ones depicted in Table 1 (unless specified otherwise), and it involves 6 different timescales:

**Table 1. Network parameters.**

| Parameter | Description | Value |
|---|---|---|
| $r_{max}$ | Maximal firing rate | 1 |
| $r_0$ | Baseline firing rate | 0 |
| $b$ | Slope parameter of the sigmoidal transfer function | 100 |
| $\theta_0$ | Base firing threshold in the absence of firing | 0.15 |
| $D_\theta$ | Constant determining the adaptation strength | 1 |
| $C$ | Constant of the Gaussian noise | 0.006 |
| $\mu$ | Mean of the Gaussian noise | 0 |
| $\sigma$ | Standard deviation of the Gaussian noise | 1 |
| $\eta$ | Learning rate | 1 |
| $\beta$ | Forgetting rate | 0.0025 |
| $\alpha_w$ | Synaptic normalization constant | 1 |
| $\alpha_r$ | Divisive normalization constant | 2 |
| $w_{max}$ | Maximal synaptic weight | 0.3 |
| $w_{min}$ | Minimal synaptic weight | -0.05 |
| $\gamma$ | Fraction of directly stimulated neurons | 0.1 |
| $N$ | Network size | 100 |
| $I_0$ | Intensity of the external stimulation | 1 |

i. $\tau_r = 1$ *arbitrary unit* (*a.u.*) is the timescale determining the neuronal activation;

ii. $T = 5$ *a.u.* is the duration of the external stimulation;

iii. $\tau_\theta = 7$ *a.u.* is the neural adaptation time constant. It is slower than $\tau_r$ so that a neuron is switched off only after having enough time to be activated;

iv. $T_{LR}$ is the length of the window adopted for calculating the learning rule's running average. The standard value is $T_{LR} = 15$ *a.u.* which was chosen to be longer than the stimulus duration $T$;

v. $\frac{1}{f_{max}} = 30$ *a.u.* is the period corresponding to the maximum stimulation frequency. It was chosen to avoid temporal overlaps between consecutive stimulations;

vi. $\tau_w = 50$ *a.u.* is the timescale determining the learning.

## Experimental paradigms

We ran different simulations to show memory formation, reinforcement and forgetting in a recurrent attractor network.

For all paradigms: i) we used a rectangular pulse train stimulation with a current of duration $T = 5$ *a.u.* and intensity $I_0 = 1$; ii) we waited for 50000 *a.u.* to give the network enough time to reach a stable standard deviation of its synaptic weights before starting the stimulation; iii) all connections and rates were initially set to 0; iv) in all cases, following the stimulation phase, there was a post-stimulation phase lasting 10000 *a.u.*, which was used to monitor the firing activity of the network in the post-stimulation period and to characterize the change of the synaptic weights in absence of any further stimulation; v) in all the plots $t = 0$ corresponds to the start of the stimulation phase.

Furthermore, to quantify the formation and evolution of the assemblies, the network properties at different points of the stimulations were tested using brief stimulation pulses ($I_0 = 1$ and $T = 1$ *a.u.*), while switching off the learning rule and the noise (i.e. in the next iteration the simulation continues as if the test input was not applied), so that the different test pulses would not interfere with the evolution of the simulations. A neuron $i$ was considered to be activated by the test stimulus if $r_i > 0.5$ at $t > 2$ *a.u.* after the presentation of the stimulus.

## Scalability of the model

We built our model using a small network of $N = 100$ neurons and assemblies representing memory patterns with initial size of 10 neurons (and, thus, a sparseness $\gamma = 0.1$). However, our system can be used with different network sizes and values of $\gamma$, with some adjustments detailed below.

If $N$ is increased but $N \cdot \gamma$ (i.e. the size of the assemblies during initial stimulation) is fixed, the parameters adopted throughout this work give qualitatively similar results.

If $N \cdot \gamma$ is varied, some parameters needs to be adjusted due to the following considerations:

i. The fraction of directly stimulated neurons changes. Consequently, the recurrent input of an assembly neuron from the rest of the assembly varies. To compensate a change from $N \cdot \gamma$ to $N' \cdot \gamma'$, $w_{max}$ (i.e. the maximum strength of the connections within the assembly) needs to be adjusted to a value $w_{max}' = \frac{N \cdot \gamma}{N' \cdot \gamma'} \cdot w_{max}$, where $w_{max}$, $N$ and $\gamma$ are the values reported in Table 1.

ii. There is a change in the minimal value of the mean connection from a non-stimulated to a stimulated neuron that has to be reached in order to recruit the non-stimulated neuron

into the assembly. In order to compensate a change from $N \cdot \gamma$ to $N' \cdot \gamma'$, $C$ needs to be adjusted to a value $C' = \frac{N \cdot \gamma}{N' \cdot \gamma'} \cdot C$, where $C$, $N$ and $\gamma$ are the values reported in Table 1.

We tested the robustness of our model using four different combinations of $N$ and $N \cdot \gamma$: $N = 500$ and $N \cdot \gamma = 10$; $N = 500$ and $N \cdot \gamma = 50$; $N = 1000$ and $N \cdot \gamma = 10$; $N = 1000$ and $N \cdot \gamma = 50$.

It should be noted that, in case the number of assemblies alternatively stimulated is changed, the stimulation frequencies used should also change in order to avoid stimulating different assemblies at the same time. Consequently, the forgetting rate β might have to be adjusted for an effect in the change of assembly sizes upon stimulation frequency to be observed.

We tested the robustness of our model using six different combinations of "number of assemblies" and $N \cdot \gamma$ in networks of $N = 1000$: i) 2 assemblies with $N \cdot \gamma = 10$; ii) 2 assemblies with $N \cdot \gamma = 20$; iii) 10 assemblies with $N \cdot \gamma = 10$; iv) 10 assemblies with $N \cdot \gamma = 20$; v) 20 assemblies with $N \cdot \gamma = 10$; vi) 20 assemblies with $N \cdot \gamma = 20$. Since the maximum "number of assemblies" tested across these simulations was 20 and the minimum distance between consecutive stimulations to avoid temporal overlaps was 30 a.u. (see the previous section "Parameter choices"), we used $f_1 = \frac{1}{600 \ a.u.}$ as the highest frequency. Given the use of lower frequencies compared to the rest of the work, we used $\beta' = \beta/_{20}$, where β is the forgetting rate indicated in Table 1.

## Supporting information

**S1 Text. Supplementary text for "A dynamic attractor network model of memory formation, reinforcement and forgetting".**
(PDF)

**S1 Fig. Assembly evolution upon stimulus repetition until convergence.** Starting at time 0 a. u., 10 neurons were stimulated for 70000 times with $f = 1/(60 \ a.u.)$. Bottom: Number of assembly neurons over time. Inset, top left: zoom with enlarged time scale. Top, right: firing rate for all neurons at the time of the 70000[th] stimulation (the 10 directly stimulated neurons are on top).
(TIF)

**S2 Fig. Firing rate for all neurons and weight matrix at the time of the 100000[th] stimulation.** Starting at time 0 a.u., 10 neurons were stimulated for 100000 times with $f = 1/(30 \ a.u.)$. The firing rate plot and the weight matrix show that almost all the network had been recruited into one assembly after 100000 stimulations.
(TIF)

**S3 Fig. Assembly evolution with two concurrent patterns stimulated at the same frequency until convergence.** Starting at time 0 a.u., two non-overlapping populations of 10 neurons each were stimulated at different times with $f = 1/(60 \ a.u.)$ for 70000 times. Bottom: number of neurons per assembly over time. Inset, top left: zoom with enlarged time scale. Top, right: weight matrix at the end of the stimulation paradigm.
(TIF)

**S4 Fig. Connections among stimulated assemblies within networks of N = 1000 with different combinations of number of assemblies and number of stimulated neurons.**
A-B-C-D-E-F) Connectivity, at t = 350000 a.u., among stimulated assemblies within networks of 1000 neurons, in case of different number of assemblies and number of stimulated neurons, namely: A) 2 assemblies of 10 stimulated neurons ($N \cdot \gamma = 10$); B) 2 assemblies of 20 stimulated

neurons; C) 10 assemblies of 10 stimulated neurons; D) 10 assemblies of 20 stimulated neurons; E) 20 assemblies of 10 stimulated neurons; F) 20 assemblies of 20 stimulated neurons. In each simulation, two stimulation frequencies ($f_1 = 1/(600\ a.u.)$ and $f_2 = 1/(1200\ a.u.)$) were used, with half of the assemblies stimulated with $f_1$ and half of the assemblies stimulated with $f_2$. For better visualization, only connections in one direction are shown (however, reciprocal connections have the same value). Connections among the non-stimulated neurons are not shown. In none of the simulations we observed the formation of overlaps between different assemblies. (Parameters: $\beta = 0.000125$.).
(TIF)

**S5 Fig. Model behaviour following selective removal of its stability mechanisms.** A) Model without hard bounds of the synaptic weights. A specific population of 10 neurons was stimulated repeatedly at a repetition frequency $f = \frac{1}{30\ a.u.}$. The mean weight of the network increased without limit until the stimulated neurons would not return to baseline activation after the end of the external stimulation. The weight successively decreased, due to forgetting and due to the fact that sustained co-activation for longer than $T_{LR}$ (i.e. length of the window adopted for calculating the learning rule's running average; see Table 1 in the main text) does not result in learning (see Eq 8 in the main text). B) Model without weight decay. The mean and standard deviation of all network connections in absence of any external stimulation are displayed in the case of model without forgetting. Without forgetting, the standard deviation kept increasing, even if the other stability mechanisms were implemented. (Parameters: $\beta = 0$). C) Model without synaptic normalization: A specific population of 10 neurons was stimulated repeatedly at a repetition frequency $f = \frac{1}{30\ a.u.}$. The firing rates for all network neurons at different times are displayed. In absence of synaptic normalization, the formation of new connections with new recruited neurons lead to more prolonged assembly activation, eventually leading to uncontrolled network activity. (Parameters: $\alpha_w = 0$; $\alpha_r = 2$). D) Model without divisive normalization. A specific population of 10 neurons was stimulated repeatedly at a repetition frequency $f = \frac{1}{30\ a.u.}$. The firing rates for all network neurons are displayed, showing that, in absence of divisive normalization, the formation of new connections with new recruited neurons lead to uncontrolled network activity. (Parameters: $\alpha_w = 2$; $\alpha_r = 0$).
(TIF)

**S6 Fig. Comparison of assembly evolution for patterns stimulated with the same frequency in different experimental paradigms.** A) Number of neurons per assembly over time in case of "Single-pattern networks", "Network with 2 patterns and 1 frequency", "Networks with 2 patterns and 2 frequencies". B) Number of neurons per assembly at different times for each of two stimulation frequencies ($f = \frac{1}{60\ a.u.}$; $= \frac{1}{120\ a.u.}$) in three different experimental paradigms. It should be noted that the assemblies' names "P1" and "P2" correspond, for each paradigm, to the ones used in the figures of the main text (Figs 6A, 7A and 8A).
(TIF)

## Author Contributions

**Conceptualization:** Marta Boscaglia, Rodrigo Quian Quiroga.

**Data curation:** Marta Boscaglia.

**Formal analysis:** Marta Boscaglia.

**Funding acquisition:** Rodrigo Quian Quiroga.

**Investigation:** Marta Boscaglia, Rodrigo Quian Quiroga.

**Methodology:** Marta Boscaglia, Chiara Gastaldi, Wulfram Gerstner, Rodrigo Quian Quiroga.

**Software:** Marta Boscaglia, Chiara Gastaldi.

**Supervision:** Rodrigo Quian Quiroga.

**Validation:** Marta Boscaglia, Chiara Gastaldi, Wulfram Gerstner, Rodrigo Quian Quiroga.

**Visualization:** Marta Boscaglia.

**Writing – original draft:** Marta Boscaglia, Rodrigo Quian Quiroga.

**Writing – review & editing:** Marta Boscaglia, Chiara Gastaldi, Wulfram Gerstner, Rodrigo Quian Quiroga.

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
