## [Decision Letter · Decision Letter 0]

13 May 2023

Dear Miss Boscaglia,

Thank you very much for submitting your manuscript "A dynamic attractor network model of memory formation, reinforcement and forgetting" for consideration at PLOS Computational Biology.

As with all papers reviewed by the journal, your manuscript was reviewed by members of the editorial board and by several independent reviewers. In light of the reviews (below this email), we would like to invite the resubmission of a significantly-revised version that takes into account the reviewers' comments.

We cannot make any decision about publication until we have seen the revised manuscript and your response to the reviewers' comments. Your revised manuscript is also likely to be sent to reviewers for further evaluation.

Sincerely,

Marcus Kaiser, Ph.D.

Academic Editor

PLOS Computational Biology

Marieke van Vugt

Section Editor

PLOS Computational Biology

Reviewer's Responses to Questions

**Comments to the Authors:**

Reviewer #1: Boscaglia et al. designed a Hebbian network with an online learning rule, random background activity, and biologically plausible dynamics to model the phenomena that familiar memories recruit more neurons while inactive memories are gradually forgotten. I found the paper well-written with an easy-to-follow exposition. The model design choices are minimal and reasonable, including the adaptive firing rate threshold, weight decay-based forgetting, and normalization rules. The results are clearly explained in terms of the model design. I have no essential concerns regarding the technical soundness of the model or the results.

On the other hand, it is not entirely clear what conceptual insights or experimentally testable hypotheses the proposed model offers. The model is explicitly designed to capture dynamic aspects of memory beyond the remit of standard Hebbian networks and, as such, may be seen as a proof-of-concept of the authors' working hypothesis. Beyond this, the in silico results are descriptive and confirmatory in nature. Many results, such as forgetting, are direct consequences of the model design. Others, such as the enlargement of memory ensembles by recruiting randomly co-firing neurons or the orthogonality between different memories, are not entirely trivial yet still closely tied to specific design choices (noisy background activity and "synaptic normalization," respectively), as the authors also highlighted. It is not clear what experimentally testable hypothesis the model proposes, nor did the authors put forward any. Some model properties seem already incompatible with experimental results: not all memories are orthogonal, and the same hippocampal neuron may respond to multiple stimuli (the authors acknowledge this in the Discussion as needing further study). A model need not capture every experimental detail, but it should offer non-obvious insights or inform future experiments.

The authors' main working hypothesis is that familiar concepts have larger memory ensembles due to background firing activity that happens to co-occur with the stimulus. An alternative hypothesis, which seems to me conceptually distinct, is that a frequent concept is experienced under more contexts than an unfamiliar one; the diverse contexts may then be incorporated into a large memory assembly through Hebbian learning. This alternative is in fact equally well described by the authors' model if "background" activity is re-interpreted to mean not just random activity, but any activity concurrent with the remembered experience. It is not obvious whether the authors intend to accommodate this interpretation (and what alternative hypotheses the model would be incompatible with).

Minor concerns

Equation 4: Is it possible for the firing rate threshold to decay asymptotically toward 0, or is it lower-bounded at theta_0? Related, when describing Fig. 4C, the authors refer to a threshold of 0.15. It is a bit confusing since 1) the threshold is dynamically changing and 2) none of the inputs in the histogram actually reach 0.15.

Page 5, last paragraph: Rather than "rather than," it seems more accurate to say "both...and".

Page 8, third paragraph: It is somewhat misleading to describe the assembly size increase as "larger" for the sparser network since the graph shows a relative size increase. Should not the absolute size increase be larger for the denser network?

Page 9, third paragraph, fourth sentence: The sentence is very confusing. Do you mean that the orthogonal memories modeled here must only correspond to non-associated memory items?

Fig. 7A is described at some length. However, is it not trivial due to the symmetry between the two patterns? The only difference is that pattern 2 always lags pattern 1 by t=30, which is expected to be a negligible difference.

Fig. 7C, right: the x-axis labels are too closely spaced and thus a bit confusing

Reviewer #2: In this intriguing paper, the authors investigate the dynamics of memory encoding in the ecologically-relevant scenario in which different stimuli are encountered with different probabilities (per unit time).

In the standard attractor model, as commonly used to model memory storage and retrieval, the neurons encoding for the different memories (asymptotically) are chosen randomly with a fixed probability and the resulting memories are embedded with uniform strength into the synaptic matrix. This is, clearly, an oversimplified description, which, in fact, does not appear to be fully consistent with experimental observations in the medial temporal lobe (as discussed at length in the manuscript).

To study how memory encoding depends on the history of the stimulation, the authors numerically investigate a model network that features several physiologically-grounded plasticity mechanisms. The dynamics of the model network, under different protocols of stimulation, are carefully dissected.

The authors report several interesting, and in my opinion important, results. In particular: (i) the number of neurons encoding for a particular stimulus increases as a function of the frequency with which the stimulus is encountered; (ii) neurons in an "expanding" neuronal representation are recruited among non-responsive neurons, so that initially non-overlapping representations will remain non-overlapping; (iii) neurons belonging to "forgotten" memories will be automatically made available again to participate in other memory representations.

The study addresses a key issue in memory function (i.e., how memories are dynamically updated) and presents novel and important results that significantly add to the credibility/plausibility of the attractor framework. The paper is clearly written and figures are very informative. I do not hesitate in recommending publication.

Reviewer #3: The paper addresses the empirical observation that human memory representations are larger for familiar ones than unfamiliar items. The authors hypothesize that familiar representations undergo multiple repeated reactivations. This causes neurons firing with background rates, outside the original memory assembly, to form strong synaptic connections with the assembly, through ongoing Hebb plasticity rule. Several mechanisms are invoked to stabilize the system and prevent runaway of assembly sizes, connectivity strength and neuronal activity. The paper incorporates these hypotheses in a detailed recurrent network model and demonstrates through computer simulations the working of the hypothesized mechanism for the growth of memory assemblies.

The main hypothesis of the paper, as outlined above, is reasonable. However, the paper suffers from major limitations.

1. The conclusions are based on the outcome of computer experiments of the network model, with virtually no analysis. It is thus, very difficult to assess the generality, robustness and scalability of the proposed mechanism, and many questions remain.

For instance, it is unclear what governs the decay time of the different assemblies, beyond the displayed trajectories of the assemblies. Naturally, the results should be cast in the form of histograms of assembly sizes and life times derived from large scale simulations. Scalability is demonstrated by using two sizes (and the same number of assemblies). No attempt is made to scale the number of memories with network sizes.

2. Notably, most of the results are for very long time in arbitrary units or number of repetitions. No attempt is made to translate to biological or behavioral time scales. Are the number of repetitions reasonable?

3. The model adopts multiple stability mechanisms (hard bound on synaptic weights, neuronal divisive normalization, synaptic divisive normalization) and weight decay. Are all these necessary for stabilizing the system?

4. The paper completely ignores the question of retrieval. Are these assemblies dynamic attractors? What would be the retrieval mechanisms and how it will be affected by the enormous imbalance between assembly sizes? What do the neurons in the assembly code? In most memory models, each assembly code for high resolution information embedded in the pattern, but as the assembly grows in size in a stochastic manner it is unclear what their activation during recall stands for?

5. Relatedly, the assumed very extensive repeated stimulation of memory patterns as a reasonable model of familiar memories. It seems more plausible that at least in part, reactivation of memories is a result of an internal process which is self-supporting. This process has been studied in the literature (for instance, Lansner et al., Kreiman and Shaham) and seems to be ignored here. Thus, large assemblies can be the result of only a small imbalance in external rehearsal which then drives a self-sustained process of spontaneous reactivation.

In conclusion, the work suffers from crucial deficiencies which severely limit the interest in the reported results.

**Have the authors made all data and (if applicable) computational code underlying the findings in their manuscript fully available?**

Reviewer #1: **No: **The code is promised to be available at the time of publication but not yet available at the time of this review.

Reviewer #2: Yes

Reviewer #3: Yes

PLOS authors have the option to publish the peer review history of their article (what does this mean?). If published, this will include your full peer review and any attached files.

Reviewer #1: No

Reviewer #2: No

Reviewer #3: No
---

## [Decision Letter · Decision Letter 1]

7 Nov 2023

Dear Miss Boscaglia,

Thank you very much for submitting your manuscript "A dynamic attractor network model of memory formation, reinforcement and forgetting" for consideration at PLOS Computational Biology. As with all papers reviewed by the journal, your manuscript was reviewed by members of the editorial board and by several independent reviewers. The reviewers appreciated the attention to an important topic. Based on the reviews, we are likely to accept this manuscript for publication, providing that you modify the manuscript according to the review recommendations.

Sincerely,

Marcus Kaiser, Ph.D.

Academic Editor

PLOS Computational Biology

Marieke van Vugt

Section Editor

PLOS Computational Biology

Reviewer's Responses to Questions

**Comments to the Authors:**

Reviewer #1: As in my last review, I have no concerns with the soundness of the study. This evaluation does not change since the authors have made no new claims.

At the same time, my previous concern remains. I believe the study a) contributes a proof-of-concept model replicating experimental results but b) primarily describes the model (cf. biology). My reading of the authors’ reply is that they acknowledge this characterization.

The journal serves both computational and biology audiences, so I think an ideal paper should use a computational model to derive a new biological insight or suggest a novel experiment. Therefore, I asked about a testable hypothesis for future experiments. I am not sure the authors fully addressed this request. The authors reply,

'[…] based on these results we predict that the repeated presentation of novel stimuli (e.g. an unknown person) should lead to the formation of gradually larger assemblies, which, due to the sparse sampling of single neuron recordings (and of the assemblies responding to the stimuli), should give an increasing tendency to find single neuron responses to novel stimuli with stimulus repetition.'

The reply seems to refer to the same hypothesis that motivated the model in the first place, not a novel prediction the model has uniquely allowed the authors to make. Indeed, the authors present the hypothesis as motivating the model, not vice versa.

This concern does not take away from the value of the study in offering a quantitative, working model of a simple hypothesis regarding a relevant question. Again, I appreciate the clean, minimal model and the paper’s clear exposition. Meanwhile, I believe the revision has not increased the significance of the study to the discipline.

The authors can potentially improve the significance of the study by committing to specific modeling choices as biological hypotheses, thus turning the model properties resulting from those choices into testable predictions. The onus is on the authors to find predictions that are experimentally testable. For example, here are some non-trivial predictions one may derive from the model, but they are not easy to test:

1. Background activity is required to enlarge memory assemblies. This prediction seems difficult to test without a method to manipulate background activity.

2. Synaptic normalization (S_{w,i}) is required to keep memory assemblies orthogonal. This hypothesis also seems challenging to test. Is synaptic normalization biologically plausible? If so, is there an experimental way to manipulate (e.g., remove) synaptic normalization?

Reviewer 3 commented on the many stimulus repeats required to establish an assembly and the need to relate model time units to a realistic time scale. I understand the general difficulty of linking model parameters to physical units. I also appreciate that physical correspondences can be out-of-scope for a particular computational model (e.g., people do not expect deep neural networks to model membrane voltage dynamics). Nevertheless, if the authors could commit to an interpretation, that would add specificity to the model predictions and, thus, the paper’s conceptual relevance.

I believe the writing can benefit from more clearly delineating the properties of the model per se and the properties the authors expect to generalize to biology. For example, the abstract states,

'Specifically, we show that a dynamic interplay between Hebbian learning and background firing activity determines the relationship between the memory assembly sizes and their frequency of stimulation.'

I found this claim uncomfortable as it seems to draw a general conclusion, but the result is entirely about the model. In fact, the claim is partly tautological with the modeling assumptions (‘Here we develop a modeling approach to provide a mechanistic hypothesis of how hippocampal neural assemblies evolve differently, depending on the frequency of presentation of the stimuli.’), so it adds little new beyond the proof-of-concept (i.e., the model ‘works’). Try changing the word ‘determines’ to ‘can explain’—the two ways of phrasing may illustrate the difference between committing to a biological hypothesis vs. designing a model to explain an observation.

A similar comment applies to the remainder of the claims in the abstract: assemblies remain independent; unstimulated connections are forgotten. Each is a direct consequence of a modeling choice (synaptic normalization; weight decay). Because modeling choices are (in principle) alterable, what should we learn from the study beyond the properties of the specific model?

Minor suggestions

- The first two sentences in the abstract seem to make a logical leap. Why are larger representations necessarily easier to recall?

- Fig. 2A: The text first describes the two matrices on the right before those on the left. Could the authors arrange the figure and the text in the same order, to aid the reader?

- Figs. 2A and 4C invite comparisons. Why does Fig. 4C (no connections beyond neurons 1–10 at t_10) differ from the right side of Fig. 2A (some connections beyond neurons 1—10 at t_15)? I guess it is because Fig. 4C had pauses between stimulations. However, Fig. 4A suggests to me that Fig. 4 used continuous stimulation. It would help to clarify how Figs. 2A and 4 differ.

- Fig. 7A: Is this result expected based on symmetry? The authors reply that there could be a winner-take-all dynamic. What is the specific hypothesis—that P1 (the one slightly leading by t = 30 a.u.) will always win over P2? If the winner is stochastic from run to run, the t-test would still not be informative.

- Figs. 7A, 8A: It would help to plot the conditions from Fig. 6A as a reference. I would expect that, e.g., two f=1/60 patterns competing for neurons in Fig. 7A would slow down assembly growth vs. Fig. 6A. It would be nice to be able to do this comparison.

- ‘Relatively larger’: The authors made this change to address a previous minor suggestion, but I think it can be clearer. How about ‘proportionally larger’? I am not sure about the interpretation that this difference is because ‘as in this case there were more neurons that could be recruited into the pattern.’ Again, the number of neurons recruited into the denser pattern is likely higher. The result could just be because the denominator is larger for the denser pattern.

- The Results section does not mention heterosynaptic normalization, even though it is featured prominently in the abstract. I suggest the authors introduce this model design when first describing the orthogonality result (Fig. 7).

- There is a typo in ‘may incorporates’

Reviewer #2: The quality and clarity of the manuscript, already very high in my opinion, have further improved with the present revision.

**Have the authors made all data and (if applicable) computational code underlying the findings in their manuscript fully available?**

Reviewer #1: **No: **Cannot evaluate because the code is not yet available

Reviewer #2: None

PLOS authors have the option to publish the peer review history of their article (what does this mean?). If published, this will include your full peer review and any attached files.

Reviewer #1: No

Reviewer #2: No

Figure Files:

Data Requirements:

Reproducibility:

References:

---

## [Editor Report · Decision Letter 2]

2 Dec 2023

Dear Miss Boscaglia,

We are pleased to inform you that your manuscript 'A dynamic attractor network model of memory formation, reinforcement and forgetting' has been provisionally accepted for publication in PLOS Computational Biology.

Best regards,

Marcus Kaiser, Ph.D.

Academic Editor

PLOS Computational Biology

Marieke van Vugt

Section Editor

PLOS Computational Biology

---

## [Editor Report · Acceptance letter]

15 Dec 2023

PCOMPBIOL-D-23-00556R2 

A dynamic attractor network model of memory formation, reinforcement and forgetting

Dear Dr Boscaglia,

I am pleased to inform you that your manuscript has been formally accepted for publication in PLOS Computational Biology. Your manuscript is now with our production department and you will be notified of the publication date in due course.

With kind regards,

Zsofi Zombor
